# A study paradigm integrating prospective epidemiologic cohorts and electronic health records to identify disease biomarkers

Jonathan D. Mosley et al.[#]

Defining the full spectrum of human disease associated with a biomarker is necessary to advance the biomarker into clinical practice. We hypothesize that associating biomarker measurements with electronic health record (EHR) populations based on shared genetic architectures would establish the clinical epidemiology of the biomarker. We use Bayesian sparse linear mixed modeling to calculate SNP weightings for 53 biomarkers from the Atherosclerosis Risk in Communities study. We use the SNP weightings to computed predicted biomarker values in an EHR population and test associations with 1139 diagnoses. Here we report 116 associations meeting a Bonferroni level of significance. A false discovery rate (FDR)-based significance threshold reveals more known and undescribed associations across a broad range of biomarkers, including biometric measures, plasma proteins and metabolites, functional assays, and behaviors. We confirm an inverse association between LDL-cholesterol level and septicemia risk in an independent epidemiological cohort. This approach efficiently discovers biomarker-disease associations.

Biomarkers are reproducible measures of a physiological state. When associated with disease risk, biomarkers can facilitate early diagnosis or risk stratification, and in instances where the biomarker is a mediator of disease, can be targeted to prevent or treat disease[1]. Defining the complete spectrum of disease outcomes associated with a biomarker not only provides insights into disease mechanisms, but may also reveal potential beneficial and adverse effects of modulating biomarker levels. Traditionally, disease biomarkers are identified and characterized using epidemiological study designs, which directly measure the biomarker and outcomes in the same individuals. A limitation of these studies is that they often assess a only single outcome, ascertained over years or decades. Defining the extended set of phenomic associations requires measuring the biomarker in very large populations comprising large numbers of clinical outcomes, which typically is not feasible. Efficient, cost-effective approaches that quickly and comprehensively define the clinical epidemiology of putative biomarkers are needed.

Electronic health record (EHR) data resources could be suitable for biomarker discovery and characterization due to the presence of diverse outcomes with large sample sizes. However, the existing data are restricted to measurements which have proven clinical value. Hence, newly discovered or nonclinical biomarkers are not available for clinical and epidemiological characterization in EHRs. More recently, EHR data sets have been linked to DNA biobanks, thereby creating resources comprising many individuals with both dense clinical and genetic data[2,3]. This has enabled study designs such as phenome wide association study (PheWAS), which serially tests associations between a variable and a large collection of clinical diagnoses extracted from an EHR data set[4,5].

Leveraging genetic information across multiple studies can bypass limitations of biomarker studies in single populations. A genetic predictor based on common single nucleotide polymorphisms (SNPs) can capture the genetic component of variability in a given biomarker. This predictor can then be used to compute a genetically predicted level of the biomarker into any genotyped population. Importantly, this genetically predicted level can be used to test for epidemiological associations with potential diseases whose risk is also modulated by genetic risk factors[6,7]. Thus, biomarkers measured in one genotyped population can be associated with outcomes ascertained in a second genotyped population in whom the biomarker was not measured.

Constructing a robust SNP predictor of a biomarker's level typically requires large scale genome wide association studies (GWAS) to identify SNPs that are reliably associated with the biomarker. For many unproven biomarkers, data sets sufficiently powered to enable SNP discovery by GWAS are not yet available. Alternative genetic approaches which simultaneously analyze large number of SNPs can measure the collective contribution of these SNPs to phenotype variability using relatively modest sample sizes[8–10]. Methods such as Bayesian sparse linear mixed modelling (BSLMM) have extended these approaches and can compute SNP weights across large numbers of SNPs, and these can then be used to calculate genetically predicted phenotype values[11]. By not having to identify a collection of SNPs meeting the rigid $p$ value thresholds expected from SNP discovery approaches in order to construct predictors, BSLMM overcomes limitations of relying on GWAS to identify SNPs.

We couple the BSLMM approach with PheWAS to enable a discovery-oriented study design whereby a genetic predictor of a biomarker level is developed in an initial genotyped population and then used to impute biomarker levels into a larger, deeply phenotyped population. Biomarker measurements used here are from the prospective Atherosclerosis Risk in Communities (ARIC) study[12] and the clinical population is from the Electronic Medical Records and Genomics (eMERGE) network, a consortium of medical centers with EHR-linked DNA biobanks[13]. We show that this approach identifies well-characterized clinical associations across a wide range of putative biomarkers and enables discovery of associations between biomarkers and clinical outcomes.

## Results

**Biomarker genetics and model performance.** We used BSLMM to generate genetically predicted levels for 53 biomarkers measured in 7740 subjects participating in the ARIC study (Fig. 1a and Supplementary Table 1). The underlying genetic architectures of the biomarkers varied considerably. Estimates of the additive genetic variances explained by the common SNPs (proportions of variance explained (PVE)) ranged from 0.57 (red blood cell distribution width [RDW]) to 0.03 (carotid wall thickness) (Supplementary Fig. 1). Biomarkers related to blood coagulation such as the activated partial thromboplastin time (aPTT) and Von Willebrand Factor (vWF) levels had the largest portion of the genetic risk attributable to SNPs of large effect size (PGE = 0.87 and 0.78, respectively), while phenotypes such as diastolic blood pressure, smoking and waist circumference had small contributions from large effect SNPs (PGE < 0.05 for each phenotype) (Supplementary Fig. 1).

We computed the genetically predicted level for each of the 53 biomarkers in the EHR population. The median amount of the genetic variance (PVE) explained by the genetically predicted biomarker level was 6.5% [interquartile range: 0.9–14.9%] (Supplementary Fig. 1). The frequency distributions of the predicted levels ranged from trimodal for biomarkers levels that were heavily driven by SNPs with relatively large effect sizes (e.g., vWF) to approximately normally distributed for highly polygenic phenotypes (e.g., waist circumference) (Supplementary Fig. 2). We employed PheWAS to identify clinical diagnoses associated with each predicted biomarker. There were 116 biomarker-phenotype associations among 25 biomarkers that were significant at an experiment-wide Bonferroni $p < 0.05$ (Fig. 1b and Supplementary Data 1). To ascertain the validity of the associations, we quantified how many of 42 prespecified positive control biomarker-diagnosis pairs were significantly associated. Of 42 expected associations, 21 (50%) were significantly associated with a positive-control phenotype (Fig. 2, Supplementary Fig. 3 and Supplementary Table 2). To ensure that the associations were not due to population stratification, we reran the analyses for the top associations, adjusting for 20 PCs. Only one result was significantly impacted, an association between the biomarker "RDW", a measure of the range of sizes of red blood cells, and a diagnosis of "Disorders of iron metabolism" which are diseases affecting iron levels that often manifest with abnormal red blood cell sizes (Supplementary Fig. 4). Unlike the other biomarkers, RDW was measured in a small number of subjects ($n = 1736$), which may have contributed to its outlier status.

Many of the 116 biomarker associations were with clusters of phenotypes denoting an elevated level of the biomarker or disease subtypes attributed to the biomarker (Fig. 2). For instance, systolic blood pressure (SBP) was associated with 6 hypertension-related diagnoses such as "Hypertenstpdelive chronic kidney disease" (odds-ratio [OR] = 1.15, 9% CI: [1.10–1.19], $p = 3.9 \times 10^{-12}$), which is a diagnosis of elevated blood pressure with concomitant kidney disease. There were fewer associations with known diseases attributable to downstream effects of a biomarker, such as the association between body mass index and obstructive sleep apnea (OR = 1.09 [1.06–1.13] $p = 8.1 \times 10^{-8}$).

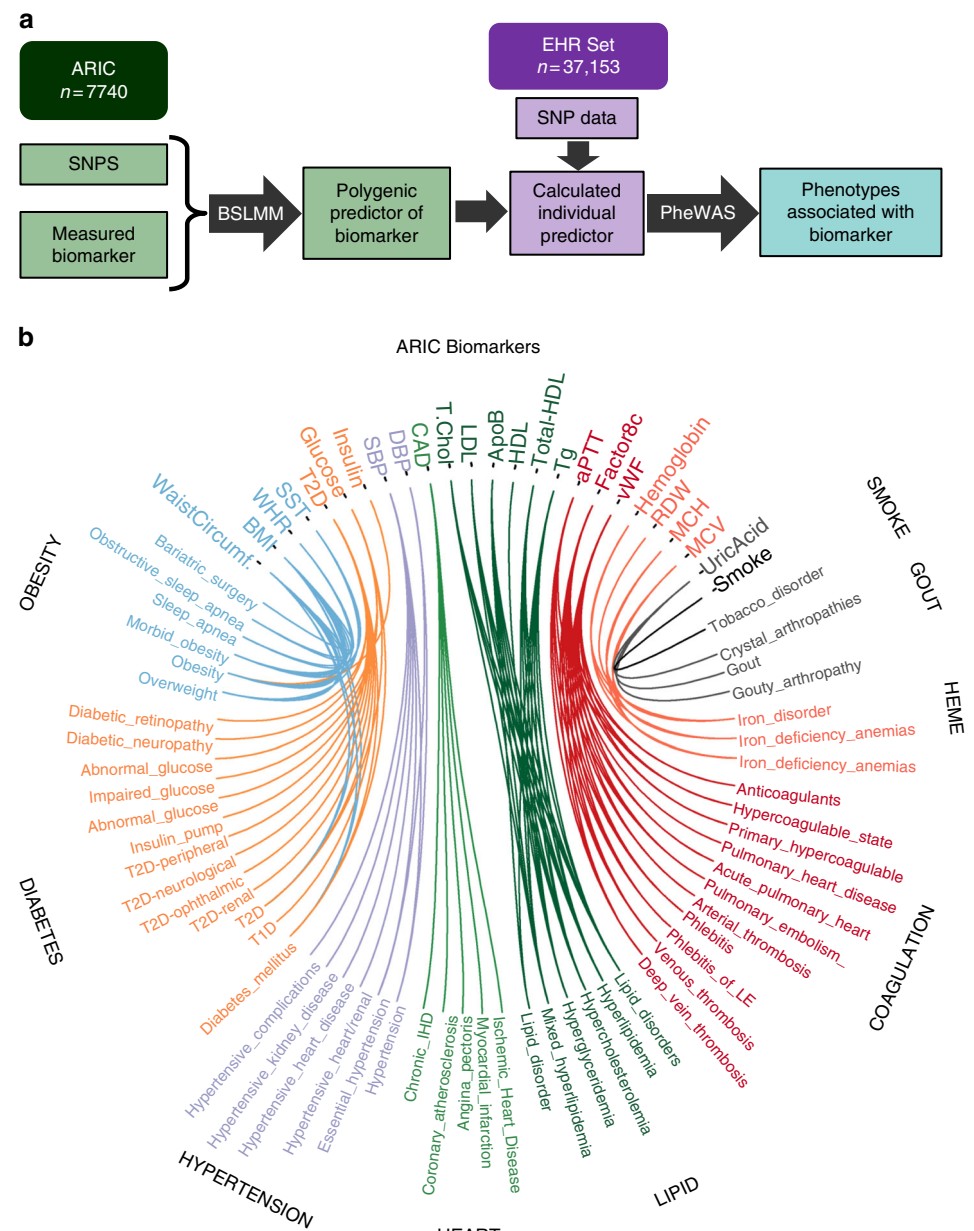

**Fig. 1** Overview. **a** Overview of the study design. Bayesian sparse linear mixed modelling (BSLMM) was used to compute SNP weights for 53 biomarkers from the ARIC study. These weights were used to compute genetically predicted biomarkers in the EHR data set and phenome wide scanning (PheWAS) was used to identify clinical phenotypes associated with the genetically predicted biomarker. **b** Circos plot showing the 116 significant associations (Bonferroni $p < 0.05$) between the genetic predictors of the ARIC biomarkers and pheWAS phenotypes. Associations are denoted by lines. Coloring is used to highlight similar groups of biomarkers and pheWAS phenotypes

**Evaluating an FDR-based selection threshold.** For 32 (76%) of the 42 positive controls pairs, the expected positive control phenotype was among the top 5 strongest associations in the PheWAS analysis (Fig. 2). For instance, the most significant association with the ARIC biomarker high-sensitivity C-reactive protein (CRP) was with "Hypertriglyceridemia" and the second was with the positive control diagnosis "Elevated CRP" ($p = 0.0009$) (Supplementary Fig. 5). We explored whether employing an false discovery rate (FDR) $p$ value selection threshold, which is often used in conjunction with PheWAS to enable discovery[5], would identify more positive control associations. Using an inclusion threshold of FDR $q < 0.1$, 28 (66.7%) of the 42 positive control pairs met the inclusion criteria (Fig. 2 and Supplementary

Table 2). Ten (24%) of the 42 biomarkers had neither significant nor highly ranked associations with a positive controls. These included serum creatinine, phosphorous, protein, potassium, hematocrit and factor VII levels, and neutrophil count.

There were 377 biomarker-phenotype associations significant at FDR $q < 0.1$ (Supplementary Fig. 6 and Supplementary Data 2). The biomarkers with the largest differences in the numbers of associations meeting a FDR threshold and not a Bonferroni threshold were SBP (26 versus 6 associations) and waist circumference (32 versus 7) (Supplementary Fig. 1). Of the 26 SBP associations, 24 clinical diagnoses associated with SBP at FDR $q < 0.1$ (Fig. 3a) and represented 6 distinct diseases know to be associated with blood pressure (hypertension, stroke, heart

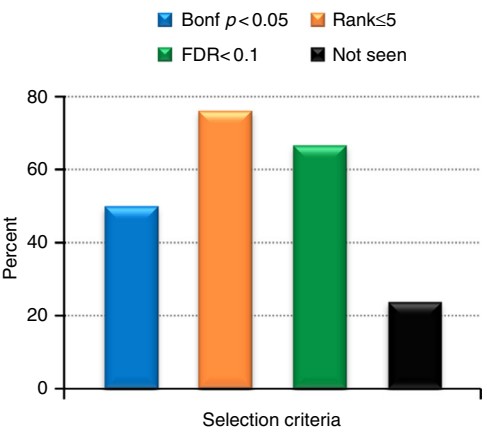

**Fig. 2** Associations with positive controls. Positive control biomarker-phenotype pairs were identified a priori for 42 ARIC biomarkers. The histogram quantifies the percentage of pairs with Bonferroni $p < 0.05$, rank order value $\leq 5$, false discovery rate (FDR) $q < 0.1$ or not seen by any of the criteria. Some pairs may fall into multiple categories

disease, peripheral vascular disease (PVD), kidney disease, and gout), and the other 2 diagnoses were for nonspecific electrolyte abnormalities (e.g., elevated potassium), which are often attributable to antihypertensive medications or late-stage sequelae of hypertensive disease (such as kidney failure) (Fig. 3b)[14–18]. The six diagnoses that also met a Bonferroni level of significance were for hypertension and hypertension subtypes (Fig. 3b). Similarly, for waist circumference (Fig. 3c), 29 of the 32 diagnoses associated at FDR $q < 0.1$ were for 11 known known obesity-related diagnoses (Fig. 3d)[19–22]. Only 3 of the 11 of the associated diseases (obesity, diabetes, and sleep apnea) were represented among diagnoses meeting a Bonferroni selection threshold (Fig. 3d). The other 3 of 32 associated diagnoses were hypotension ($n = 2$ diagnoses) and treatment with aspirin. In sum, the FDR significance threshold identified more established biologically relevant biomarker-disease associations than the Bonferroni threshold.

Waist circumference had the largest number of associations with FDR $q < 0.1$ of any biomarker. Across all associations, there was marked skewing of the ORs toward values greater than 1 (skewness = 1.40), indicating that waist circumference had weak positive associations across a large number of phenotypes (Fig. 3e and Supplementary Data 3). The phenotype with next largest skewness value was low-density lipoprotein cholesterol (LDL-C), which was skewed in the opposite direction (skewness = −1.15).

**Overview of associations**. There were numerous associations meeting the FDR threshold where the diagnosis matched the known biology or epidemiology of the biomarker. For instance, genetically predicted higher triglyceride levels were associated with multiple vascular diseases including PVD (OR = 1.10 [1.05–1.15]), ischemic heart disease (IHD) (OR = 1.06 [1.02–1.09]), abdominal aortic aneurysm (OR = 1.12, [1.06–1.17]), and renal artery atherosclerosis (OR = 1.12 [1.05–1.19]) (Fig. 4a and Supplementary Fig. 7). A genetic predictor of lifetime smoking burden (pack-years) was associated with obesity (OR = 1.05 [1.02–1.07]) and alcohol use (OR = 1.12 [1.07–1.18]) (Fig. 4b). In contrast, an alcohol predictor was associated with alcohol use, but not smoking (Supplementary Fig. 8), indicating that the alcohol predictor was more weakly associated the smoking diagnosis. Genetically

predicted FEV1/FVC levels, a spirometric measure of lung function, was associated with the obstructive lung disease emphysema [OR = 0.91 [0.87–0.95]) (Supplementary Fig. 8). A magnesium level genetic predictor was associated with two diagnoses, in opposite directions, related to the pathological precipitation of electrolytes: chondrocalcinosis (OR = 0.88 [0.82–0.95]), which is caused by calcium pyrophosphate precipitation in a joint, and kidney stones (urinary calculus; OR = 1.07 [1.03–1.11]), which are frequently caused by precipitation of calcium oxalate in the kidney (Fig. 4c). Genetically predicted calcium levels were associated with hyperparathyroidism (OR = 1.12 [1.05–1.19]) (a cause of elevated calcium) (Supplementary Fig. 9). vWF, a soluble protein involved in hemostasis[23], was associated with both venous thrombosis (e.g., deep vein thrombosis [OR = 1.24 {1.18–1.29}]) and arterial thrombosis (including stroke [OR = 1.11 {1.06–1.16}] and "Acute vascular insufficiency of intestine" [OR = 1.21 {1.09–1.34}]) (Fig. 4d). Antithrombin III, another coagulation factor, was associated with venous disease (Supplementary Fig. 10).

There were also undescribed associations present among known associations. For instance, among hematologic biomarkers: platelet counts were associated with varicose veins (OR = 0.90 [0.85–0.95]) and "poisoning by anti-infectives" (OR = 0.82 [0.73–0.91]); white blood cell counts were associated with anxiety disorders[24] (OR = 1.07 [1.04–1.10]), and chronic bronchitis[25] (OR = 1.08 [1.03–1.12]); and monocyte counts were associated with tonsillitis (OR = 0.85 [0.78–0.93]) and tendon rupture (OR = 1.17 [1.08–1.25]) (Supplementary Fig. 11). Subscapular skin-fold thickness was associated with many obesity-related diagnoses and also two adverse drug phenotypes for opiates (OR = 1.14 [1.07–1.22]) and cortical steroids (OR = 1.17 [1.08–1.27]) (Supplementary Fig. 12).

**Feasibility study in African ancestry**. There was a small number of African ancestry (AA) subjects available for analysis. Since we were insufficiently powered for discovery, we ascertained how many associations observed in the European ancestry (EA) analyses had a Bonferonni-adjusted significant association at $p < (0.05/107 = 4.67 \times 10^{-4})$ in an AA cohort. Of the 116 EA associations with a Bonferroni significance, 107 were available for testing in the AA cohort. Four associations in the AA cohort had a significant association. To ascertain whether the low replication rate could be due to low power, we compared the direction of association in the AA cohort to the direction in the EA cohort for all associations with a nominal association $p < 0.1$ in the AA cohort ($n = 64$). Among these, 64/64 (100%) had the same direction of association as the EA data set, suggesting that more associations would be replicated with a larger sample size.

**Validating an LDL-C genetic association**. Genetically predicted levels of LDL-C were inversely associated with two diagnoses related to infection: bacteremia (OR = 0.91 [0.87–0.95], FDR $p = 0.009$) and septicemia (OR = 0.93 [0.90–0.96], FDR $p = 0.01$) (Fig. 5a). LDL-C was also associated with T2D with the same direction of effect. Since infections are a common sequela of T2D, we stratified the association analysis by T2D status to see whether this eliminated the associations with the infection diagnoses. In stratified analyses, the associations were only significant among subjects with T2D versus nondiabetics (e.g., for septicemia: OR = 0.91 [0.87–0.96], p = $5.3 \times 10^{-4}$ versus OR = 0.96 [0.91–1.00], $p = 0.057$, respectively) (Fig. 5b and Supplementary Table 3). To ascertain whether these associations could be recapitulated using directly measured LDL-C levels, we tested for associations between clinically measured LDL-C levels and the infection

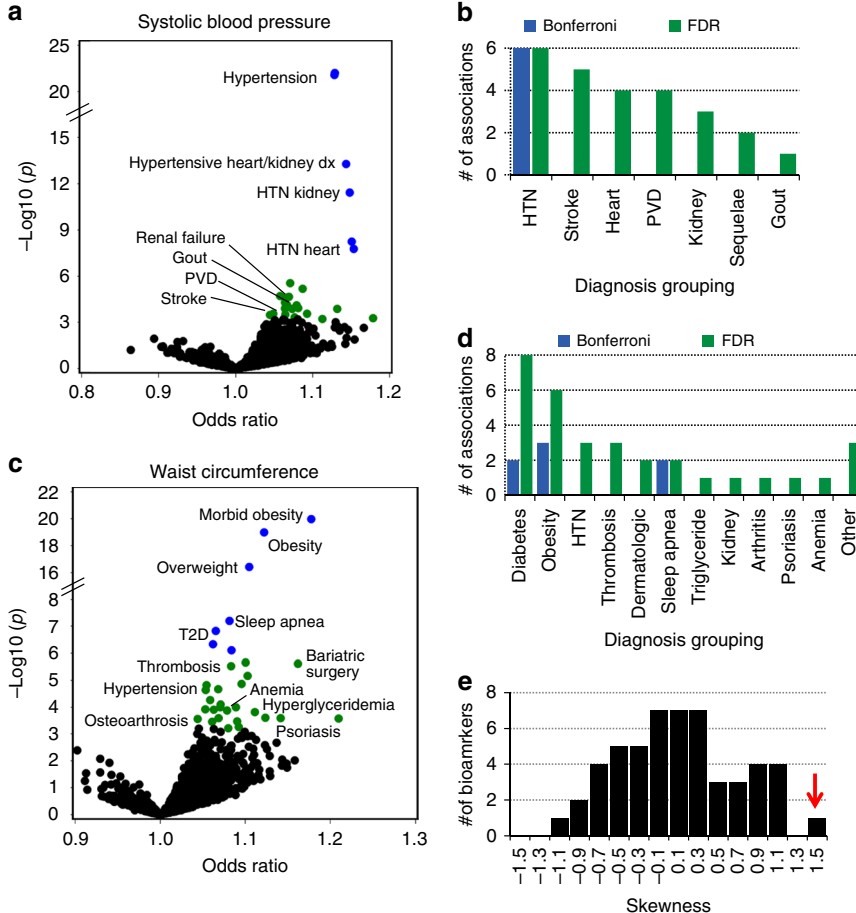

**Fig. 3** Comparison of an FDR versus Bonferroni *p* value selection threshold. **a** Scatter plot summarizing pheWAS analyses for a genetic predictor of systolic blood pressure (SBP). Each point indicates a logistic regression association analysis, adjusted for birth decade, sex, and 3 PCs, between genetically predicted waist circumference and a pheWAS phenotype. Odds ratios are per standard deviation increase in the genetic predictor. Blue and green colored circles denote associations that are significant at Bonferroni *p* < 0.05 and FDR *q* < 0.1, respectively. Only selected points are labelled for clarity. **b** Count of the number of associations, binned by disease, meeting a Bonferroni, and FDR selection thresholds. **c** PheWAS associations for a waist circumference genetic predictor and **d** count of disease associations significant by Bonferroni or FDR criteria. **e** Frequency histogram of the skewness (see Methods for calculations) of the pheWAS beta coefficients for each of the 53 biomarkers. The red arrow points to the value for waist circumference. HTN: hypertension; PVD: peripheral vascular disease; T2D: type 2 diabetes

diagnoses in an independent cohort. Specifically, we tested whether these associations could be detected by standard epidemiological methods using a previously curated cohort of 22,281 subjects who had either low (LDL-C < 60 mg/dl) and normal (LDL-C > 90 and <130 mg/dl) cholesterol levels while not taking lipid-lowering drugs. Similar to the genetic association, low measured LDL-C, as compared to normal LDL-C was associated with a decreased risk for both phenotypes (e.g., for septicemia OR = 3.54 [2.81–4.46], $p < 2 \times 10^{-16}$) (Fig. 5c and Supplementary table 4). Thus, low LDL-C, as measured by either a genetic predictor or measured directly, was associated with an increased risk of severe infection.

## Discussion
We examined 53 high quality biomarkers measured in the ARIC study and identified clinical diagnoses associated with genetic predictors of those biomarkers. We observed significant associations with genetic predictors for a broad range of biomarkers, including plasma proteins, plasma metabolites, functional assays related to clotting and lung function, electrolytes levels, and behaviors. In discovery-oriented analyses, we

identified an inverse association between genetically predicted LDL-C levels and septicemia, which was replicated in an independent EHR-derived epidemiological cohort.

Traditionally, biomarkers have been identified and validated using prospective studies. While the gold standard approach, the sample sizes and follow-up times required by prospective studies are resource-constrained, thereby limiting the number and diversity of clinical endpoints observed over time. For instance, the link between hypertension and stroke was reported by the Framingham study 17 years after the first patient was examined[26,27]. We demonstrate an approach which rapidly associates a putative biomarker with clinical outcomes based on their shared polygenic architectures. Genetics-based association approaches are effective because heritable genetic variation represents a lifelong exposure to disease risk. Thus, even modest genetic perturbations to homeostatic levels of a causative biomarker are detectable given the long duration of exposure. Our approach eliminates the requirement that biomarkers and outcomes be measured in the same populations, which allows us to significantly augment the number of disease associations that can be tested. Since these data sets can be rapidly developed and, in the case of EHR data sets, comprise a broad collection of clinically

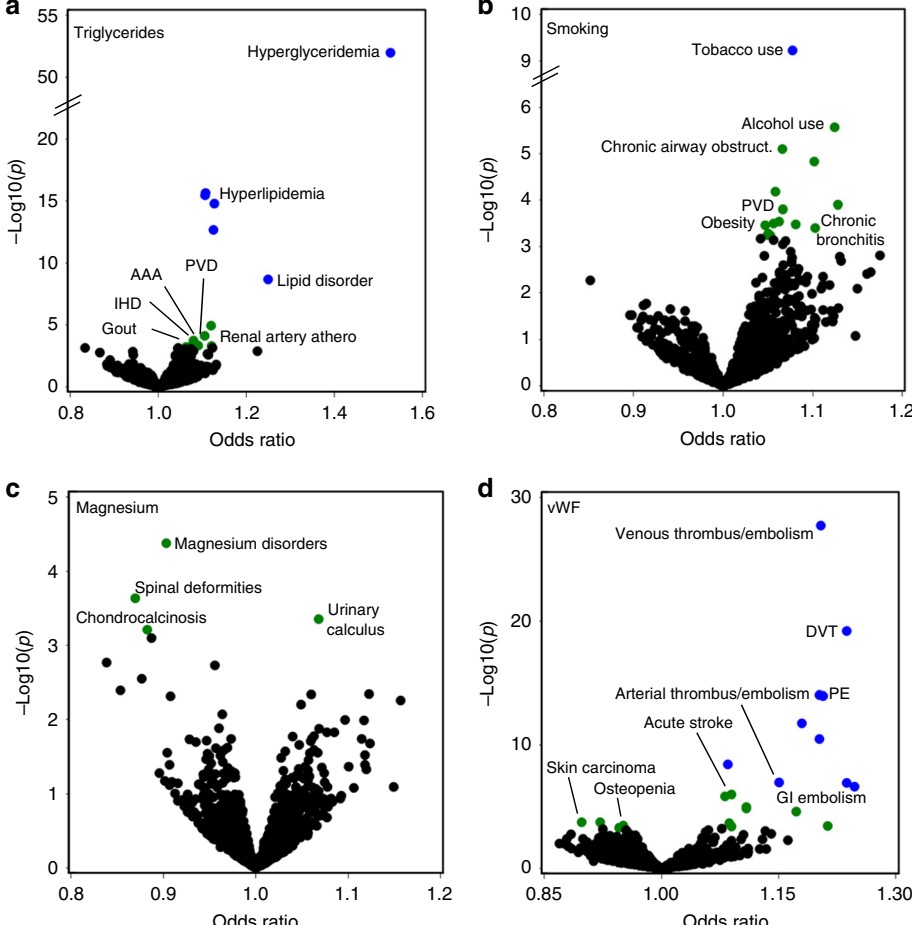

**Fig. 4** Associations for selected biomarkers. Scatter plots summarizing pheWAS analyses for genetic predictors of **a** triglyceride levels, **b** pack-years of smoking, **c** serum magnesium levels, and **d** serum Von Willebrand factor levels. Odds ratios are from logistic regression analyses, adjusting for birth decade, sex, and 3 PCs. Blue and green colored circles denote associations that are significant at Bonferroni $p < 0.05$ and FDR $q < 0.1$, respectively. PVD: peripheral vascular disease; AAA: abdominal aortic aneurysm; IHD: ischemic heart disease; DVT: deep vein thrombosis; PE: pulmonary embolism; GI: gastrointestinal

recognized outcomes, we can rapidly identify a broad spectrum of clinical disease associated with a biomarker.

The power of a SNP-based genetic predictor is optimal when it consists of only those SNPs that modulate the phenotype of interest. Thus, for phenotypes with existing GWAS conducted using large sample sizes, predictors based on the highly significantly associated SNPs perform well because they capture a modest amount of the genetic variation and the signal-to-noise ratio is high. Biomarkers without a proven clinical application, however, are more likely to be measured in relatively small data sets. Therefore, we employed an SNP weighting strategy based on linear mixed models. Mixed modelling approaches are typically more sensitive than GWAS at capturing the overall additive genetic SNP effects, especially with small sample sizes[28]. The advantages of Bayesian approaches, as used here, are that SNP weights are assigned based on an empirical estimation of the underlying genetic architecture and predictors incorporate all SNPs. These advantages enhance prediction across a range of genetic architectures and capture the contributions of SNPs with smaller effect sizes[29]. A limitation of mixed modelling approaches is that they require access to individual level data. Alternative methods, such as LD-regression[30] and LDPred[31], have been proposed to capture features of mixed models and leverage GWAS summary statistics data.

To assess the biological plausibility and validity of our approach, we examined 42 positive control associations in which the biomarker and the clinical phenotype were closely related. The positive control diagnosis was a top PheWAS association for 32 (76%) biomarker-diagnosis pairs, and was frequently the most significant association. Only 50% of the positive control associations were significant using a Bonferroni level of significance. Because the effects of a disease process can manifest across a range of phenotypes, a disease biomarker will often be associated with multiple diagnoses. In these instances, the FDR adjustment procedure is desirable since it is designed to control the family-wise error-rate in the context of positive regression dependence[5,32]. Consistently, we found that applying an FDR selection threshold identified more positive control associations (67%) and more known biological associations than a Bonferroni correction. PheWAS is typically used as a hypothesis generating tool, and the benefit of using an FDR threshold is that it identifies candidate associations while controlling for the proportion of false discoveries at a given selection threshold. Thus, at an FDR threshold of 0.1 used in these analyses, we identified far more candidate associations than when we applied a Bonferonni threshold. By design, ~10% of associations ($n = $ ~38) are expected to be false-positives, so further validation of candidate associations, as we describe for LDL-C below, is essential.

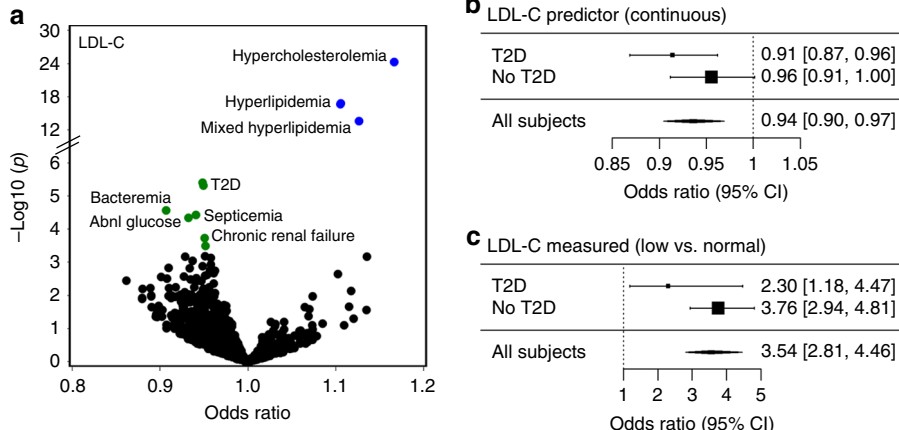

**Fig. 5** Associations with LDL cholesterol (LDL-C). **a** Scatter plot summarizing pheWAS analyses for a genetic predictor of LDL-C. Blue and green colored circles denote associations that are significant at Bonferroni $p < 0.05$ and FDR $q < 0.1$, respectively. **b** Association analysis between the LDL genetic predictor and the PheWAS septicemia phenotype, stratified by type 2 diabetes (T2D) status. Error bars represent 95% confidence intervals of odds-ratio estimates. **c** Epidemiological association between the Low (LDL-C < 60 mg/dl) versus Normal LDL-C (between 90 and 130 mg/dl) and septicemia, stratified by T2D status, using an independent EHR cohort. Odds-ratios were determined by multivariable logistic regression adjusting for age, gender and race and stratified by T2D status. Error bars represent 95% confidence intervals. T2D type 2 diabetes

For biomarkers whose levels fluctuate acutely in clinical settings, such as serum potassium levels and neutrophil counts, associations with the positive control diagnoses were often not observed. This is likely due to the fact that these diagnoses often reflect acute departures from baseline levels, rather than long-standing homeostatic variability. Some other types of positive control pairs also did not perform well, such as ankle-brachial index, a measure of blood flow to the lower extremities, and a diagnosis of PVD. Possible explanations are that (1) the clinical phenotypes are not well-ascertained (and, thus, there is high misclassification), (2) the biomarker is not measured accurately, or (3) the biomarker or the clinical phenotype does not have a strong genetic signal. Further analyses in independent data sets could clarify these possibilities.

Many of the biomarkers that we evaluated are known cardiovascular risk factors. This allowed us to demonstrate that we could rediscover known epidemiologically significant associations. We found associations in which the biomarker is either a mediator or direct contributor to disease risk, such as the associations between SBP and diagnoses of kidney disease, heart disease, and stroke[33]. SBP was also associated with gout, which is caused by precipitation of uric acid. Pharmacologically lowering uric acid levels can improve blood pressure[18]. Waist circumference had associations with many clinical diagnoses, consistent with epidemiological observations that obesity predisposes to a broad range of morbidities[34]. Among these was psoriasis, and weight loss through either dietary or surgical approaches has been shown to markedly attenuate psoriasis severity[20].

We also identified patterns of associations indicative of shared regulatory mechanisms among diseases and the biomarker. A genetic predictor of serum magnesium levels was associated with two disorders of calcium: chondrocalcinosis and kidney stones, implicating a link between magnesium and calcium regulation. Studies of Mendelian diseases point to shared mechanisms of coregulation for these factors. For instance, the disease Familial hypomagnesemia with hypercalciuria and nephrocalcinosis is caused by mutations in *claudin-16*, which leads to magnesium and calcium wasting and kidney stones[35]. The translational utility of establishing an epidemiological association between magnesium and calcium homeostasis is evidenced by current clinical practice standards, which include the evaluation and correction of

magnesium deficits for the treatment of hypocalcemia[36]. We also observed associations between predicted pack-years of smoking and diagnoses of tobacco-use, alcohol-use and obesity, suggesting that the genetic predictor of smoking behavior is capturing addiction or reward mechanisms related to eating and drinking behaviors[37]. These findings suggest that treatments, which are effective for treating one of these behaviors may also be effective for the others.

We observed a number of undescribed associations that met the FDR criteria. For instance, there was an inverse association between tonsillitis risk and a predictor of monocyte counts, a subset of white blood cells. It has been previously shown that the ratio of peripheral lymphocytes to WBCs can differentiate between tonsillitis caused by mononucleosis and tonsillitis caused by bacteremia[38]. Hence, the monocyte count biomarker may be capturing a similar pattern of changes to the monocyte cellular fraction which reflect a genetically mediated predisposition toward infection. We also observed associations with pharmacogenomic phenotypes. Genetically predicted platelet levels were inversely associated with a diagnosis of "poisoning by other anti-infectives". Thrombocytopenia (low platelets) is a complication associated with a range of medications including antibiotics, NSAIDS and anticoagulants[39,40]. Possible mechanisms accounting for the association we observed are that some antibiotics may modulate genetic mechanisms regulating platelet homeostasis or that individuals genetically predisposed to lower levels of platelets at baseline are more likely to drop below the clinically defined threshold defining thrombocytopenia. We also found that subscapular skin-fold thickness was inversely associated with adverse reactions to narcotics and adrenal steroids. The effects of steroids on skin, including inducing purpura and lipodystropy (thinning of the skin), are well described. However, a mechanism relating adverse effects of opiates to skin phenotypes is not clear.

There were a number of biomarkers for which both higher and lower predicted levels were associated with clinical diagnoses. The clinical relevance of this pattern of association is that it provides insights into potential adverse effects that may be associated with modulating the biomarker. For instance, higher levels of predicted LDL-C were associated with a diagnosis of hyperlipidemia and lower levels were associated with increased T2D risk, an

association that has been previously reported[41]. Consistent with these findings, lowering cholesterol using a statin drug is associated with an increased risk of T2D[42]. This inverse genetic relationship was also manifest in the skewness analysis, which showed that the LDL-C predictor, which demonstrated the strongest negative skewing, had a skewness value of similar magnitude, but opposite direction, as a T2D predictor (−1.15 versus 0.98).

Because a genetic predictor only captures a small portion of the phenotypic variability, it cannot substitute for the directly measured value of the biomarker in a clinical setting. We observed that predicted LDL-C levels were also inversely associated with septicemia risk. Since LDL-C measurements are frequently available and easily measured in clinical settings, we evaluated the plausibility of this candidate association by ascertaining whether clinically measured LDL-C levels were associated with infection risk. We confirmed the genetic association by showing that low levels of measured LDL-C were associated with diagnoses of bacteremia and septicemia. The epidemiological association was stronger than the genetic association and was not suggestive of an interaction with T2D status. The stronger association observed with direct measurement suggests that both the genetic and environmental modulators of LDL-C variability impact septicemia risk similarly. Thus, a direct measurement of LDL-C, which captures both genetic and environmental risk, would be expected to be more predictive than a proxy that only captures genetic risk. A similar association was seen in the prospective REGARDS population cohort ($n = 30,239$ subjects), in which it was found that individuals with lower baseline LDL-C levels had the highest risk of incident sepsis[43]. While a mechanism accounting for these findings is not known, the authors note that LDL-C is involved in clearance of bacterial toxins, as lipid-based pathogens bind to proteins which are transferred to LDL-C particles and are ultimately taken up by the liver[44]. Hence, our findings provide independent support that LDL-C level is a biomarker of infection risk.

There are several limitations to this study. PheWAS phenotypes are derived from EHR billing codes rather than a systematic ascertainment of a diagnosis. This can lead to misclassification, especially among the controls, which can attenuate the strength of associations. While the genetic predictors were designed to capture the variation of the ARIC biomarkers, an association between the classifier and diagnosis does not necessarily indicate that phenotypic variation in the biomarker is associated with the diagnosis, as it is possible that SNPs predictors are tagging variants that predispose to the clinical diagnosis through different genetic mechanisms[45]. Some biomarkers may not have a large genetic component, preventing the use of this approach. The Benjamini–Hochberg (B–H) FDR method used in these analyses may have an erroneous type 1 error rate if the pattern of associations violates the positive regression dependence assumption. We did not have sufficient subjects to fully validate our approach in other ancestries. We performed a feasibility analysis using available non-EA subjects and demonstrate that the approach recapitulates some expected results in AA populations. Further validation with large sample sizes is needed to more fully define its utility across ancestries. These analyses do not meet the criteria of a Mendelian Randomization experiment, a subclass of genetic association studies which seek to establish a causal association between a biomarker and an outcome[7,46]. Thus, similar to an epidemiological or genetic correlation study, we also do not claim causality for the associations that we report. Ultimately, prospective studies will be needed to validate associations, such as that between LDL levels and infection, which this methodology identifies.

In summary, we developed polygenic SNP predictors for biomarkers measured in the ARIC prospective study and associated

these with clinical diagnoses derived from a large EHR data set. We were able to efficiently recapitulate known epidemiological associations and identify additional associations. We anticipate this study design will become an important mechanism to rapidly identify clinical outcomes that may be associated with putative biomarkers, which will enhance the translation of these biomarkers into clinical practice.

## Methods

**Study populations.** The ARIC population comprises 13,113 genotyped adult subjects participating in the NHLBI-funded Atherosclerosis Risk in Communities longitudinal study designed to investigate the natural history of cardiovascular and atherosclerotic diseases[12]. Study subjects were recruited between 1987 and 1989 from four U.S. communities: Minneapolis, MN, Washington County, MD, Forsyth County, NC, and Jackson, MS. Genetic and phenotypic data were obtained from dbGaP (phs000280.v3.p1). For the primary analyses, only unrelated ARIC subjects of EA, defined as having >90% probability of being in the HapMap CEU cluster using STRUCTURE[47] in conjunction with ancestry informative markers, were analyzed ($n = 7740$).

The primary EHR population was derived from the eMERGE Phase I and II Network ($n = 16,923$), a consortium of medical centers using EHRs as a tool for genomic research, and from Vanderbilt University Medical Center's (VUMC) BioVU resource ($n = 20,230$) (Supplementary Table 5)[13,48]. The participating eMERGE sites were Geisinger Health System, VUMC, Marshfield Clinic, Northwestern University, Mayo Clinic and Kaiser Permanente/University of Washington, Seattle. BioVU is a deidentified collection of patients whose DNA was extracted from discarded blood and linked to phenotypes through a deidentified EHR[49]. The additional BioVU subjects were not part of eMERGE and had been previously genotyped. All subjects were born prior to 1990 and fell within four standard deviations for each of the first two principal components based on common SNPs for the subset of subjects self-identified as "White, non-Hispanic". Principal component analyses visualizing the ARIC and EHR populations with respect to HAPMAP populations are shown in Supplementary Fig. 13.

The eMERGE study was approved by the Institutional Review Board (IRB) at each site, including VUMC's IRB[13,49].

**Genetic data.** SNP genotype data were acquired on the Illumina Human660W-Quadv1_A, HumanOmni1-Quad, HumanOmni5-Quad, MEGA-EX, Human610, Human550, HumanOmniExpressExome-8v1.2A, and Affymetrix 6.0 SNP array platforms (Supplementary Table 6). Quality control (QC) steps for the EHR data sets included reconciling strand flips, verifying that allele frequencies were concordant among data sets, and identifying duplicate and related individuals[50,51]. QC for the ARIC data set followed the guidelines accompanying the dbGaP release, including removing SNPs with chromosomal anomalies and with >5 discordant calls in replicate samples and using a predefined subset of unrelated subjects. QC analyses used PLINK v 1.90β3[52]. Prior to imputation, data sets were standardized using the HRC-1000G-check tool v4.2.5 (http://www.well.ox.ac.uk/wrayner/tools/). SNPs were then pre-phased using SHAPEIT[53], imputed using IMPUTE2[54] in conjunction with the 10/2014 release of the 1000 Genomes cosmopolitan reference haplotypes. All platforms were imputed to the 1000G standard which comprises ~80,000,000 common and rare SNPs. Imputed data were filtered for a sample missingness rate <2%, a SNP missingness rate <4% and a SNP deviation from Hardy–Weinberg < 0.0001. The final analytic data set comprised a LD-reduced ($r$-square > 0.9) set of 739,681SNPs with MAF > 1% present on all platforms (the EHR data sets had 7,218,081SNPs, the ARIC data set had 7,254,201 SNPs and their intersection had 5,701,931 SNPs). Principal components were generated using the SNPRelate package[55].

**Biomarker and phenotype data.** In the ARIC data set, 53 biomarkers representing a range of laboratory, biometric, and other measurements were used to construct SNP-based predictors (shown in Supplementary Data 4). In order to assess whether genetic predictors based on the ARIC biomarkers appropriately associated with similar clinical diagnoses in the EHR data set, one or more positive control clinical diagnoses were identified for 42 of the ARIC biomarkers (Supplementary Table 2). For 25 of these positive controls, the ARIC biomarker represented the disease-defining continuous measure for a clinical diagnosis (e.g., hypertension is defined as a SBP > 140 mmHg) and for the other 17, the ARIC biomarker was linked to the clinical diagnoses based on its known biology (e.g., aPTT as a potential biomarker for diagnoses of thromboses). For a number of biomarkers, multiple closely-related clinical diagnoses were selected as potential positive controls.

EHR clinical phenotypes were based on phecodes (https://phewas.mc.vanderbilt.edu/), which are collections of related ICD-9-CM (International Classification of Disease, Ninth revision) diagnosis codes[4,5]. For each phenotype, cases are subjects with one or more instances of the code appearing in their medical record. Any eMERGE site which had fewer than 10 cases for a given phenotype was excluded for that phenotype. Phenotypes that only affected a single gender were not included in these analyses. There were 1139 clinical phenotypes with ≥300 cases in the EHR data set that were used in the PheWAS

analyses. Controls were subjects without the clinical phenotype or any closely related phecodes and whose decade of birth fell within the range of birth decades observed among cases.

**Statistics**. BSLMM, as implemented in the GEMMA v0.95α package[56], was used to construct SNP classifiers for each of the 53 ARIC biomarkers. BSLMM employs a hybrid of generalized linear mixed modelling and sparse regression models[11] and estimates the proportion of variance explained by a set of SNPs as well as the distribution of effect sizes for the SNPs. It then jointly models the contribution of all SNPs to the observed phenotypic variance. Parameters estimated by the modelling approach that are reported here are the proportion of additive phenotype variance explained by all SNPs (PVE), proportion of genetic variance explained by SNPs with relatively larger effect sizes (PGE) and the estimated number of large-effect SNPs modulating the phenotype. For each ARIC biomarker, 100,000 sampling steps were run and the parameter estimates reported are the median from the last 50,000 iterations[57]. The posterior SNP weight estimates generated by this approach were used to compute genetically predicted values for each of the ARIC biomarkers. To estimate how much of the genetic variance is explained by a biomarker predictor, a subset of 500 ARIC subjects was randomly removed and the BSLMM model was fit using the remaining subjects. The genetically predicted level of the biomarker was estimated in the 500 removed subjects, and the variance of the measured phenotype that was explained by the predicted level was calculated. The %PVE is defined as (variance explained by the biomarker/PVE)*100.

To compute the genetically predicted values for each biomarker within the EHR data set, each biomarker phenotype was first adjusted for age, sex, and two PCs using linear regression. BSLMM was then used to generate SNP weights (w) using the regression residuals. For each SNP, BSLMM computes both a small polygenic effect ($\alpha$), a large effect ($\beta$) and a posterior probability that the SNP is in the large effect group ($\gamma$) based on the underlying genetic architecture for the phenotype, as determined by the Bayesian algorithm. The SNP weight is computed using the equation: $w = \alpha + \beta\gamma$. These weights were used to compute a genetically predicted value of the ARIC biomarker for each individual in the EHR data set taking the sum of the SNP genotype (coded as 0, 1, or 2) multiplied by its corresponding weight across all SNPs:

$$\text{Predicted phenotype} = \sum_{i=1}^{\#\text{SNPs}} \left( w_i \times [\text{SNP genotype}]_i \right) \quad (1)$$

A PheWAS analysis was then performed using each genetically predicted biomarker and serially testing its association with each PheWAS phenotype using multivariable logistic regression, adjusting for three PCs, birth decade, sex, eMERGE site, and genotyping platform. The genetically predicted phenotypes values were standardized to have a standard deviation of 1, so ORs reflect the risk per standard deviation (s.d.) increase in the genetically predicted biomarker value (which is not equivalent to a s.d. increase in the original biomarker). Association analyses used SAS v9.3 (SAS Institute, Cary, NC). To adjust for multiple testing, we applied a strict Bonferroni correction and associations with $p < (0.05 < [53 \times 1139] = \sim 8.28 \times 10^{-7})$ were considered significant. In discovery-oriented analysis, we examined associations with aa B–H FDR[58] $q$ value < 0.1 (a threshold often applied to PheWAS analyses[5]).

Some PheWAS analyses appeared to yield more positive than negative (or vice versa) associations. In order to measure the extent to which the logistic regression coefficients (Beta) from a pheWAS analysis were positively or negatively skewed away from 0 (corresponding to an OR of 1), a skewness statistic was computed using the equation:

$$\text{Skewness} = \left(\sum (\text{Beta})^3 / n\right) / \left(\sum (\text{Beta})^2 / n\right)^{3/2}$$

where $n$ = the number of pheWAS phenotypes

and Beta is the regression coefficient.

Since this analysis was used to ascertain the extent to which genetically predicted phenotypes had nonsignificant associations with the PheWAS phenotypes, the analyses included additional pheWAS phenotypes for which there was low power to detect an association due to low case counts (>150 cases, $n = 1318$ total phenotypes). To eliminate the influence of strong associations, associations with FDR $q < 0.1$ were excluded from the analysis.

**Feasibility study in AA**. Both the ARIC and EHR data sets contained an insufficient number of subjects of non-EA to conduct a well-powered analysis using those populations. However, we did assess the feasibility of this approach in other ancestries by conducting a parallel analysis in self-reported black subjects and ascertained how many of the significant associations identified in the EA population were observed in non-EA population had a Bonferroni-adjusted association $p < 0.05$. ARIC and eMERGE subjects who fell within four standard deviations of the first two PCs for EHR subjects self-identified as "Black" were used. There were 2703 unrelated ARIC and 8552 eMERGE subjects (Supplementary Table 1) available for analysis. For these analyses, eMERGE data were imputed using the Michigan Imputation Server (HRC v1.1)[59]. This reference panel is enriched for individuals of EA, and also includes the diverse ancestries from the 1000 Genomes populations ($n = 2495$)[60]. Data were available for 52 of the 53 ARIC biomarkers. Due to smaller number of subjects in the EHR data set, PheWAS diagnoses with >100 cases (rather than >300) were analyzed. While 52 ARIC biomarkers were analyzed, for 19 of 52 biomarkers, the BSLMM model was not able to optimize the hyper-parameters (likely due to small sample sizes which can affect its performance). There were nine significant diagnosis-biomarker associations observed in the EA analysis for which the diagnosis was not available in this cohort.

**Epidemiological replication**. As described, we identified an association between genetically predicted LDL cholesterol levels and infection phenotypes. To determine whether this finding could be recapitulated using directly measured LDL-C levels, we used a cohort of 28,753 subjects previously extracted from VUMC's deidentified EHR to test for an epidemiological association between low LDL-C cholesterol levels and the PheWAS diagnoses for "Bacteremia" (phecode 38.3) and "Septicemia" (phecode 38). All subjects had one or more outpatient LDL-C measurements that were not taken while the patient was on a lipid-lowering medication. For subjects with multiple LDL-C measurements, the median of all measurements was used in the analyses. "Low LDL-C" subjects had a median LDL-C < 60 mg/dl and "Normal LDL-C" subjects had a median value between 90 and 130 mg/dl. Analyses were limited to subjects born before 1990 and excluded 766 subjects who were included in the genetic association analyses. After exclusions, there were 22,281 subjects available for analysis. Age, gender, self-reported race and type 2 diabetes (T2D) case/control status based on Phe-WAS codes were available for each subject (Supplementary Table 7). Logistic regression analyses were used to test the association between the infection diagnoses and LDL-C (low versus normal). Analyses were stratified by T2D status and adjusted for age, gender, and race. A $p$ value < 0.05 was considered statistically significant.

**Data availability**. All ARIC data are available through dbGaP (phs000280.v3.p1). Much of the eMERGE data presented here is also available through dbGaP (phs000360.v3.p1), and the remaining is under preparation for deposition.

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

## Acknowledgements

The authors thank the staff and participants of the ARIC study for their contributions. The authors wish to acknowledge the expert technical support of the VANTAGE and VANGARD core facilities, supported in part by the Vanderbilt-Ingram Cancer Center and Vanderbilt Vision Center. This work was supported by a career development award from the Vanderbilt Faculty Research Scholars Fund (J.D.M.), American Heart Association (16FTF30130005) (J.D.M.), PGRN (P50-GM115305), R01 GM10945, R01 LM010685, R01 HL133786-01A1 (W.W.), R01 GM120523 (Q.F.), 16SDG29090005 (J.H.K.). SLV has support through Burroughs Wellcome Fund IRSA 1015006 and a CTSA award KL2TR000446 (NCATS/NIH). VUMC's BioVU projects are supported by numerous sources: institutional funding, private agencies, and federal grants. These include the NIH funded Shared Instrumentation Grant S10RR025141; CTSA grants UL1TR002243, UL1TR000445, and UL1RR024975. Genomic data are also supported by investigator-led projects that include U01HG004798, R01NS032830, RC2GM092618, P50GM115305, U01HG006378, U19HL065962, R01HD074711; and other funding sources listed at https://victr.vanderbilt.edu/pub/biovu/?sid=229. The eMERGE Network was initiated and funded by NHGRI through the following grants: U01HG006830 (Children's Hospital of Philadelphia); U01HG006389 (Essentia Institute of Rural Health, Marshfield Clinic Research Foundation and Pennsylvania State University); U01HG006382 (Geisinger Clinic); U01HG006375 (Kaiser Permanente/University of Washington, Seattle); U01HG006379 (Mayo Clinic); U01HG006380 (Icahn School of Medicine at Mount Sinai); U01HG006388 (Northwestern University); U01HG006378; U01HG8685 (Brigham and Women's Hospital); U01HG8672 (Vanderbilt University Medical Center); and U01HG006385 (Vanderbilt University Medical Center serving as the Coordinating Center); U01HG004438 (CIDR) and U01HG004424 (the Broad Institute) serving as Genotyping Centers. ARIC is supported by NHLBI contracts (HHSN268201100005C, HHSN268201100006C, HHSN268201100007C, HHSN268201100008C, HHSN268201100009C, HHSN268201100010C, HHSN268201100011C, and HHSN268201100012C). ARIC/GENEVA was supported by NHGRI grant U01HG004402 (E. Boerwinkle).

## Author contributions

J.D.M., Q.F., Q.S.W., D.M.R., J.C.D., and C.M.S. contributed to study design. J.D.M., Q. F., C.M.S., S.L.V., C.M.S., D.R.C., and J.H.K. contributed to data analysis and interpretation. T.L.E., L.B., W.Q.W., L.K.D., C.A.M., W.T., C.G.C., G.P.J., A.S.G., M.R.P., D.R. C., E.B.L., D.S.C., I.J.K., J.A.P., P.L.P., M.H.B., J.G.L., B.N., M.S.W., M.D.R., K.M.B., S.S.

V., S.T.W., J.C.D., and D.M.R. contributed to data acquisition. J.D.M., Q.F., Q.S.W., S.L. V., T.J.W., C.M.S., J.C.D., and D.M.R. contributed to manuscript preparation.

**Additional information**

**Competing interests:** The authors declare no competing interests.

Jonathan D. Mosley[1,2], QiPing Feng[1], Quinn S. Wells[1], Sara L. Van Driest [1,3], Christian M. Shaffer[1], Todd L. Edwards [4], Lisa Bastarache[2], Wei-Qi Wei[2], Lea K. Davis [1], Catherine A. McCarty[5], Will Thompson[6], Christopher G. Chute[7], Gail P. Jarvik[8], Adam S. Gordon[8], Melody R. Palmer[8], David R. Crosslin[9], Eric B. Larson[8,10], David S. Carrell[10], Iftikhar J. Kullo[11], Jennifer A. Pacheco [12], Peggy L. Peissig[13], Murray H. Brilliant[14], James G. Linneman[13], Bahram Namjou[15], Marc S. Williams[16], Marylyn D. Ritchie [17], Kenneth M. Borthwick[17], Shefali S. Verma[17], Jason H. Karnes[18], Scott T. Weiss[19], Thomas J. Wang[2], C. Michael Stein[1], Josh C. Denny[1,2] & Dan M. Roden[1,2,20]

[1]Department of Medicine, Vanderbilt University Medical Center, Nashville, TN 37232, USA. [2]Biomedical Informatics, Vanderbilt University Medical Center, Nashville, TN 37232, USA. [3]Department of Pediatrics, Vanderbilt University Medical Center, Nashville, TN 37232, USA. [4]Vanderbilt Epidemiology Center, Vanderbilt University Medical Center, Nashville, TN 37232, USA. [5]Essentia Institute of Rural Health, Duluth, MN 55805, USA. [6]Department of Medicine, Feinberg School of Medicine, Northwestern University, Chicago, IL 60611, USA. [7]Schools of Medicine, Public Health, and Nursing, Johns Hopkins University, Baltimore, MD 21205, USA. [8]Department of Medicine (Medical Genetics), University of Washington, Seattle, WA 98195, USA. [9]Departments of Biomedical Informatics and Medical Education, University of Washington, Seattle, WA 98195, USA. [10]Kaiser Permanente Washington Health Research Institute, Seattle, WA 98101, USA. [11]Department of Cardiovascular Diseases, Mayo Clinic, Rochester, MN 55905, USA. [12]Center for Genetic Medicine, Northwestern University Feinberg School of Medicine, Chicago, IL 60611, USA. [13]Biomedical Informatics Research Center, Marshfield Clinic Research Institute, Marshfield, WI 54449, USA. [14]Center for Human Genetics, Marshfield Clinic Research Institute, Marshfield, WI 54449, USA. [15]Center for Autoimmune Genomics and Etiology, Cincinnati Children's Hospital Medical Center, Cincinnati, OH 45229, USA. [16]Genomic Medicine Institute, Geisinger Health System, Danville, PA 17822, USA. [17]Biomedical and Translational Informatics, Geisinger Health System, Danville, PA 17822, USA. [18]Department of Pharmacy Practice and Science, University of Arizona College of Pharmacy, Tucson, Arizona 85721, USA. [19]Channing Division of Network Medicine, Department of Medicine, Brigham and Women's Hospital, Harvard Medical School, Boston, MA 02115, USA. [20]Department of Pharmacology, Vanderbilt University Medical Center, Nashville, TN 37232, USA

