## [Peer Review file · Nature Communications]

Reviewers' comments:

Reviewer #1 (Remarks to the Author):

The authors outline an approach which rapidly associates a putative biomarker with clinical outcomes based on their shared polygenic architectures. The core benefit is that doing so eliminates the requirement that biomarkers and outcomes be measured in the same populations. Since these linked data sets can be rapidly developed and comprise a broad collection of clinically-recognized outcomes, it is possible to identify a broad spectrum of clinical disease associated with a biomarker.

The good:

- 1) Neat example of data integration
- 2) Find a new biomarker for septicemia, showing that the method works

The not so good:

- 1) How well the model works is not clearly quantified

With such a strategy, a key point would be defining how well the BSLMM predicted the biomarker. The proportion of variance explained is shown in Supp table 1, but how accurate the predicted value was – with a few concrete examples - is not described. It is necessary to know the uncertainty in the biomarker prediction from genetics. (i.e. The reverse of the extent to which genetics explain the biomarker.)

Additionally, not all biomarkers are created equal, and some have much less flexibility for variance than others. For example, blood pH has an extremely strong correlation with coma and death, and so is tightly regulated. A highly accurate prediction from this model would be absolutely necessary for this biomarker. This is in contrast to waist circumference, where a margin of error of the prediction would be acceptable. This is one reason why a description of accuracy is essential.

- 2) Another question is the usefulness of predicting biomarkers: why not simply use the SNP to predict the phenotype that the biomarker is serving as a marker for?

As a specific example, the authors determine that LDL-C, ('Validating an LDL genetic association', last paragraph in the results) as predicted by SNP data, inversely correlates with septicemia.

Why bother determine the predicted LDL-C, when one could simply use the SNP data? The association is very cool and potentially useful, however and could probably be learned directly from the EHR data where both LDL-C and Septicemia would be present.

Overall, it is unclear why we need to join prospective dataset with EHR datasets using the "shared genetic architecture" as the join. In the discussion, the authors mention that biomarkers have been 'identified and validated using prospective studies. While the gold standard approach, the sample sizes and follow-up times required by prospective studies are resource-constrained, thereby limiting the number and diversity of clinical endpoints observed over time.' The massive medical record system allows for multiple endpoints to be observed in a huge sample size.

Then, why not just use the patient data, and leave out the SNPs? It would be cool to see an example how if only patient data was used, this (LDL-C septicemia) association could not be seen. Doing so would show how the SNPs (and joining over the genetic data) is necessary.

Detailed comments:

Intro > Paragraph 3:

I found this confusing, needing further explanation. Especially "By not having to identify specific SNPs to use as predictors, this approach overcomes limitations of relying on GWAS to identify SNPs."

Intro > Paragraph 4:

I think "estimation" or "prediction" should be used instead of "imputation". (My understanding is that there is no biomarker data for the EHR population. So they are predicting this, rather than filling in missing data.)

Methods > Genetic Data:

Could the authors include what fraction of genetic data needed imputation?

Methods > Statistics:

In the Predicted phenotype formula: there is a parenthesis ")" missing. Could the authors explicitly explain what is the sum over?

Results > Validating an LDL genetic association:

I found it confusing and disconcerting that the SNP-predicted LDL-C association is barely significant once stratified by T2D state, but then is much stronger in the epidemiological study (measured-biomarker associating with the phenotype). Could the authors explain this better? Do we expect the association to be stronger in the epidemiological study, since genetics are only a poor indicator of biomarker status?

(Note: Figure 5 is truncated)

Figure 2:

I don't think it's particularly useful to format the plot as a 2x2 grid, and the uneven gridlines are confusing. I would prefer that the caption make it clear that yellow dots are those with FDR $q > 0.1$ but were among the top 5.

Supplementary Figure 2:

What are the units of the x-axis? And could the authors explain the titles (e.g. "p = Insulin")? It seems Supplementary Figure 2 and 3 are the same figure split to fit on two pages. It would be better to keep it a single figure displayed over two pages.

Supplementary Table 7:

To calculate the Odds Ratio, I believe the data could also be broken into "have/don't have the biomarker". Could the authors provide that data? If this isn't how the Odds Ratio was calculated, then I think it needs more explanation.

Reviewer #2 (Remarks to the Author):

In this paper, the authors used a linear mixed model approach to estimate the joint effects of all SNPs on a biomarker in the ARIC cohort, and then applied the estimated SNP effects to create a genetic predictor of the biomarker in a validation cohort with electronic health record in order to assess the associations of the genetic predictor of the biomarker with a number of diagnoses. While the analytical paradigm is useful and some of the results are interesting, there are several major concerns need to be addressed.

Population stratification

Since the authors used all the common SNPs to create the genetic predictor for a biomarker, any subtle population stratification can be picked up by the SNPs, which could potentially lead to false-

positive associations of the predictor with the diagnoses if the validation cohort is also stratified. Although the first 2 PCs as covariates were fitted in the linear mixed model analysis to estimate the SNP effects and the first 3 PCs were fitted in the logistic regression analysis to assess the biomarker-diagnosis associations, the first 2 or 3 PCs might not be sufficient to control for population stratification in a highly stratified cohort such as ARIC. It is important to know whether the results hold when the first 20 PCs are fitted in both analyses.

False discovery rate

To correct for multiple testing, the authors used a false discovery rate (FDR) of 0.1 to define significant biomarker-diagnosis associations. This means that ~38 of the 377 "significant" associations are expected to be false positives given a FDR of 0.1. Such a high number of false positives could lead to wrong conclusions about the genetic relationships between biomarkers and diagnoses. I would use a much more stringent FDR (or type-I error) threshold so that the number of false positives is smaller than 1 in the whole experiment.

Because population stratification is a potential confounding factor, the authors might need to visualize the population structure of the cohorts used in their analyses together with the European/African samples from the 1000 Genomes Project.

The total number of SNPs after HRC/1000G imputation should be >7 million. The authors reported that after LD reduction with a LD r-squared threshold of 0.9, there were ~740K SNPs. This number seems too small to me. Please clarify.

Page 5. The equation used to compute the genetic predictor is not clear. Gamma is not defined. It seems that all the SNPs have an effect of alpha but only a subset of SNPs have an effect of beta with gamma being the mixing probability. The authors need to clearly define all the variables and parameters in the equation.

Page 5. I believe that the beta in the equation to compute skewness is not the same as the one used to compute the genetic predictor above. Please use a different notation.

Page 5. "from 0 (corresponding to a log(odds-ratio) of 1)" odds-ratio of 1?

Page 6. "For these analyses, eMERGE data were imputed using the Michigan Imputation Server". Did the authors mean that non-EA samples were imputed to 1000G using the Michigan Imputation Server? There is no non-European sample in HRC.

Page 7. "an increased waist circumference increased the risk of the clinical diagnosis". This sentence reads as if the association is causal.

Page 7. "The phenotype with next largest skewness value was LDL cholesterol, which was skewed in the opposite direction (skewness=-1.15)." It seems that LDL cholesterol is protective against most diagnoses. The authors might need to comment on this.

Page 7. "In contrast, an alcohol predictor was associated with alcohol use, but not smoking (Supplementary figure 6)." This could just be a power issue.

Page 8. "For 55 (14.5%) of the 377 significant associations, there is no clear or known genetic relationship between the biomarker and the associated diagnosis (Supplementary table 9)." It is difficult to know whether these are novel associations or false positives given the high false discovery rate.

Page 8. "we ascertained how many associations observed in the EA analyses had a nominal association at $p < 0.05$ in a AA cohort." The use of a p-value threshold of 0.05 is not justified. The authors need to apply a stringent threshold to correct for multiple testing.

Page 9. "We found a novel inverse association between genetically predicted LDL-C levels and septicemia which was replicated in an independent EHR-derived epidemiological cohort." The authors might also need to fit the first 20 PCs as covariates in the replication analysis.

The text in Figure 1B are difficult to read.

Figure 2A. It might be better to use a different color to represent the associations that fall in both "FDR<0.1" and "Rank<=5" categories.

Figure S1. "the proportion of variance proportion of genetic variance". the proportion of genetic variance?

Comments to the editor:

In addition to the comments below, we have made additional edits to manuscript (such as shortening the abstract, adding an author contribution section) to ensure that it conforms to the format of the journal. These changes are highlighted in the tracked changes document.

We are grateful for the input from the reviewers. Please find a point-by-point response to the comments they have provided.

Responses to reviewer #1:

Please note that we modified the order of some comments from reviewer #1 in order to enhance the narrative flow of our responses.

Overall, it is unclear why we need to join prospective dataset with EHR datasets using the "shared genetic architecture" as the join. In the discussion, the authors mention that biomarkers have been 'identified and validated using prospective studies. While the gold standard approach, the sample sizes and follow-up times required by prospective studies are resource-constrained, thereby limiting the number and diversity of clinical endpoints observed over time.' The massive medical record system allows for multiple endpoints to be observed in a huge sample size. Then, why not just use the patient data, and leave out the SNPs? It would be cool to see an example how if only patient data was used, this (LDL-C septicemia) association could not be seen. Doing so would show how the SNPs (and joining over the genetic data) is necessary.

The reviewer appropriately suggests that the optimal epidemiological study design is a large, population with complete ascertainment, in which it is possible to measure any novel biomarker. While EHR data sets can be proxies for this ideal study design, their utility is markedly limited because biomarkers without proven clinical utility are not measured on these populations. Hence, it is not possible to measure an association between a novel non-clinical biomarker and an EHR diagnoses. Linking EHRs to DNA biobanks provides a mechanism to construct a genetic proxy of a measured biomarker which can then be used for hypothesis testing. To ensure that we convey this point, we have revised the Introduction to include the following paragraph (Page 2, paragraph 3):

"Electronic health record (EHR) data resources could be suitable for biomarker discovery and characterization due to the presence of diverse outcomes with large sample sizes. However, the existing data are restricted to measurements which have proven clinical value. Hence, novel or non-clinical biomarkers are not available for clinical and epidemiological characterization in EHRs. More recently, EHR data sets have been linked to DNA biobanks, thereby creating resources comprising many individuals with both dense clinical and genetic data.^{2,3}"

The reviewer is correct that the LDL-septicemia association can be recapitulated using patient data, as we demonstrate. An example of a finding that we identified that cannot be seen in analysis using EHR patient data is the association between Anti-thrombin III protein levels and thrombosis. In a clinical setting, it is extremely rare that Anti-thrombin III levels are measured, so this measure is effectively unavailable in an EHR data set. Consequently, this association cannot be demonstrated in an EHR data set without the use of genetic data.

1) How well the model works is not clearly quantified. With such a strategy, a key point would be defining how well the BSLMM predicted the biomarker. The proportion of

variance explained is shown in Supp table 1, but how accurate the predicted value was – with a few concrete examples - is not described. It is necessary to know the uncertainty in the biomarker prediction from genetics. (i.e. The reverse of the extent to which genetics explain the biomarker.) Additionally, not all biomarkers are created equal, and some have much less flexibility for variance than others. For example, blood pH has an extremely strong correlation with coma and death, and so is tightly regulated. A highly accurate prediction from this model would be absolutely necessary for this biomarker. This is in contrast to waist circumference, where a margin of error of the prediction would be acceptable. This is one reason why a description of accuracy is essential.

We agree with the reviewer that the accuracy of the biomarker level is critical in order to appropriately inform clinical decision making. Consistently, we do NOT propose to use a genetic proxy for a biomarker in a clinical setting, as the genetics only account for a portion of the overall variability. Furthermore, for a biomarker like blood pH which can function as an integrated and dynamic marker of global health status, a genetic predictor would likely not be appropriate. As we explained in response to the prior comment, we are using genetic predictors as a proxy measure for the purpose of extending the capabilities of EHR data sets to enable biomarker discovery and characterization. We have modified the text in the Discussion to clarify that we do not propose that the predictor substitute for directly measured values (Page 11, paragraph 3):

“Because a genetic predictor only captures a small portion of the phenotypic variability, it cannot substitute for the directly measured value of the biomarker in a clinical setting.”

To determine how much of the genetically-mediated variability the genetic predictors capture, we reran the genetic prediction algorithm (BSLMM) using ARIC patients, but excluded 500 random individuals. We then computed the genetically predicted biomarker level in the 500 excluded individuals and compared the genetic estimates to the measured values to estimate the proportion of the variance accounted for by the genetic predictor. We show this data in Supplementary Figure 2, where we express the results as the percent of the estimated additive genetic heritability (PVE) of the biomarker. As our results demonstrate, the proportion of the genetic variance captured by the predictor was often modest, consistent with genetic studies identifying sources common SNP variation underlying phenotype variability. We have modified the Methods to describe this new analysis (Page 5, paragraph 2):

“To estimate how much of the genetic variance is explained by a biomarker predictor, a subset of 500 ARIC subjects was randomly removed and the BSLMM model was fit using the remaining subjects. The genetically predicted level of the biomarker was estimated in the 500 removed subjects, and the variance of the measured phenotype that was explained by the predicted level was calculated. The %PVE is defined as (variance explained by the biomarker/PVE)*100.”

We have updated the Results to reference the figure (Page 7, paragraph 1):

“The median amount of the genetic variance explained by the genetically predicted biomarker level was 6.5% [interquartile range: 0.9%-14.9%] (Supplementary figure 2).”

2) Another question is the usefulness of predicting biomarkers: why not simply use the SNP to predict the phenotype that the biomarker is serving as a marker for? As a

specific example, the authors determine that LDL-C, ('Validating an LDL genetic association', last paragraph in the results) as predicted by SNP data, inversely correlates with septicemia. Why bother determine the predicted LDL-C, when one could simply use the SNP data? The association is very cool and potentially useful, however and could probably be learned directly from the EHR data where both LDL-C and Septicemia would be present.

As the reviewer suggests, an alternative approach is to perform a different type of analysis, such as a genetic correlation analysis, to directly associate biomarkers with phenotypes based on SNP variation. While these approaches are powerful, they require that either summary statistics from large-scale GWAS or individual level data with large numbers of cases. We have applied these approaches to EHR data sets (for instance, see Mosley et al. *Circ Cardiovasc Genet.* 2016 Dec;9[6]:521-530), but found their performance is suboptimal for clinical phenotypes with relatively small numbers of cases. We describe these alternative approaches in the Discussion (Page 9, paragraph 4).

To clarify, what we refer to as the "genetically predicted LDL-C level" is computed solely from the SNP data. We clarify the method of computing the genetically predicted value in response to another comment from the reviewer (see below). Hence, in direct response to the reviewer's question, we are "simply using" the SNP data to make the association.

As indicated previously, the reviewer is correct that the LDL-C association could be learned directly from the EHR. Indeed, we show that association can be recapitulated using EHR data. Based on the reviewer's comment, we have modified the manuscript to further clarify that the replication of the LDL-C/septicemia association was performed using directly measured LDL-C levels. The purpose of this analysis was to demonstrate that a novel association with a genetic biomarker could be recapitulated using directly measured levels. We do not propose that SNP-based associations are superior, but they provide a mechanism to conduct a study design that is not otherwise feasible using traditional epidemiological approaches. We have modified the Methods and Results to clarify the LDL-C replication analysis:

Methods (Page 6, paragraph 3):

"As described below, we identified an association between genetically predicted LDL cholesterol levels and infection phenotypes. To determine whether this finding could be recapitulated using directly measured LDL-C levels, we used a cohort of 28,753 subjects previously extracted from VUMC's de-identified EHR to test for an epidemiological association between low LDL-C cholesterol levels and the PheWAS diagnoses for "Bacteremia" (phecode 38.3) and "Septicemia" (phecode 38)."

We have updated the Results to read, in part (Page 9, paragraph 1):

"To ascertain whether these associations could be recapitulated using directly measured LDL-C levels, we tested for associations between clinically measured LDL-C levels and the infection diagnoses in an independent cohort. Specifically, we tested whether these associations could be detected by standard epidemiological methods using a previously-curated cohort of 22,281 subjects who had either low (LDL-C<60 mg/dl) and normal (LDL-C>90 and <130 mg/dl) cholesterol levels while not taking lipid-lowering drugs."

Detailed comments:

Intro > Paragraph 3:

I found this confusing, needing further explanation. Especially “By not having to identify specific SNPs to use as predictors, this approach overcomes limitations of relying on GWAS to identify SNPs.”

We have clarified this sentence to read (Page 3, paragraph 2):

“Methods such as Bayesian sparse linear mixed modelling (BSLMM) have extended these approaches and can compute SNP weights across large numbers of SNPs, and these can then be used to calculate genetically predicted phenotype values.⁹ By not having to identify a collection of SNPs meeting rigid p-value thresholds expected from SNP discovery approaches in order to construct predictors, BSLMM overcomes limitations of relying on GWAS to identify SNPs.”

Intro > Paragraph 4:

I think “estimation” or “prediction” should be used instead of “imputation”. (My understanding is that there is no biomarker data for the EHR population. So they are predicting this, rather than filling in missing data.)

We have replaced the word “imputation” with estimation.

Methods > Genetic Data:

Could the authors include what fraction of genetic data needed imputation?

Most of the genotyping platforms comprised between 500,000-1,000,000 measured SNPs. All platforms were imputed to the 1000G standard which is composed of ~80,000,000 common and rare SNPs. Hence, overall ~98% of the SNP data is imputed. We have added a sentence to the Methods (Page 4, paragraph 3):

“All platforms were imputed to the 1000G standard which comprises ~80,000,000 common and rare SNPs.”

Methods > Statistics:

In the Predicted phenotype formula: there is a parenthesis “)” missing. Could the authors explicitly explain what is the sum over?

We have added the missing parenthesis. We have also rewritten this section of the Methods to clarify the computation (Page 5, paragraph 3):

“To compute the genetically predicted values for each biomarker within the EHR data set, each biomarker phenotype was first adjusted for age, gender and 2 PCs using linear regression. BSLMM was then used to generate SNP weights (w) using the regression residuals. For each SNP, BSLMM computes a small and large effect for the SNP, as determined by the underlying genetic model estimated by the BSLMM algorithm. The SNP weight (w) is the sum of the large and small effect sizes. These weights were used to compute a genetically predicted value of the ARIC biomarker for each individual in the EHR data set taking the sum of the SNP genotype (coded as 0, 1 or 2) multiplied by its corresponding weight across all SNPs (~740,000):

$$\text{Predicted phenotype} = \sum_{i=1}^{\#SNPs} (w_i \times [\text{SNP genotype}]_i)”$$

Results > Validating an LDL genetic association:

I found it confusing and disconcerting that the SNP-predicted LDL-C association is barely significant once stratified by T2D state, but then is much stronger in the epidemiological study (measured-biomarker associating with the phenotype). Could the

authors explain this better? Do we expect the association to be stronger in the epidemiological study, since genetics are only a poor indicator of biomarker status? (Note: Figure 5 is truncated)

The reviewer is correct that the best explanation for the stronger association seen with the directly measured LDL-C is that the genetics are a limited proxy for LDL status. The results may also suggest that environmental factors impacting both LDL and septicemia risk are more influential than the genetic factors. We have modified the Discussion to read (Page 11, paragraph 3):

“The epidemiological association was stronger than the genetic association and was not suggestive of an interaction with T2D status. The stronger association observed with direct measurement suggests that both the genetic and environmental modulators of LDL-C variability impact septicemia risk similarly. Thus, a direct measurement of LDL-C, which captures both genetic and environmental risk, would be expected to be more predictive than a proxy that only captures genetic risk.”

Figure 2:

I don't think it's particularly useful to format the plot as a 2x2 grid, and the uneven gridlines are confusing. I would prefer that the caption make it clear that yellow dots are those with FDR $q > 0.1$ but were among the top 5.

It is also possible that embedding the figure in the manuscript, as instructed, adversely affected the image. Since Figure 2A primarily displays the raw data underlying Figure 2B, we have moved Figure 2A to a new supplementary figure (Supplementary figure 5). We explored alternate formats for the figure, but found that the 2x2 grid most effectively conveyed an overview of the sensitivity and specificity of the biomarkers. We have modified the legend describing the labelling of the dots as suggested. Note that, per suggestions from Reviewer #2, we have also added additional dot colorings identifying additional partitioning of the data.

Supplementary Figure 2:

What are the units of the x-axis? And could the authors explain the titles (e.g. “p = Insulin”)? It seems Supplementary Figure 2 and 3 are the same figure split to fit on two pages. It would be better to keep it a single figure displayed over two pages.

We have modified the figure legend to clarify that the units of the x-axis are standard deviations from the mean. We have replaced the titles to clarify that each histogram corresponds to the distribution of a genetically predicted biomarker within the EHR population. We have also clarified that the figure is a display split over 2 pages. Note that this figure is now Supplementary Figure 3 in the revised manuscript.

Supplementary Table 7:

To calculate the Odds Ratio, I believe the data could also be broken into “have/don't have the biomarker”. Could the authors provide that data? If this isn't how the Odds Ratio was calculated, then I think it needs more explanation.

For this table, the odds ratio is from a logistic regression analysis where the genetically predicted biomarker value is the independent, continuous variable. The biomarker was not analyzed as a categorical variable. To clarify this, we have added the following language to the table:

“Each row reports an association between the genetically predicted biomarker (a continuous variable) and a clinical phenotype as the outcome. Odds-ratio estimates are

from a logistic regression association analysis, adjusted for birth decade, sex and 3 principal components.”

Responses to reviewer #2:

Population stratification

Since the authors used all the common SNPs to create the genetic predictor for a biomarker, any subtle population stratification can be picked up by the SNPs, which could potentially lead to false-positive associations of the predictor with the diagnoses if the validation cohort is also stratified. Although the first 2 PCs as covariates were fitted in the linear mixed model analysis to estimate the SNP effects and the first 3 PCs were fitted in the logistic regression analysis to assess the biomarker-diagnosis associations, the first 2 or 3 PCs might not be sufficient to control for population stratification in a highly stratified cohort such as ARIC. It is important to know whether the results hold when the first 20 PCs are fitted in both analyses.

The reviewer brings up an important methodological point, namely that predictors comprising a large number of SNPs could drive PheWAS associations by simply stratifying the EHR population. If extensive stratification were influencing the results, it would be expected that the genetic predictor for each biomarker would produce a similar pheWAS result, since the stratification would be fixed, regardless of the predictor. In contrast, the pheWAS results are highly specific to the biomarkers. To confirm this, we have re-run all of the analyses incorporating 20 principal components, as suggested by the reviewer. We show using scatter plots comparing the beta coefficients and p-values for the top associations from the original and fully adjusted analysis demonstrate that there is minimal impact from adjustment for all but 1 association. These results are summarized in Supplementary Figure 4. We reference this figure in the Results (Page 7, paragraph 2):

“Adjusting the analyses for up to 20 PCs only impacted one association, suggesting that the observed results were not due to population stratification (Supplementary figure 4).”

Because population stratification is a potential confounding factor, the authors might need to visualize the population structure of the cohorts used in their analyses together with the European/African samples from the 1000 Genomes Project.

We show the relationship between our populations and HAPMAP reference populations by principal component analysis in a new figure (Supplementary Figure 1). As the figure demonstrates, the individuals in both the EHR and ARIC populations cluster around the European ancestry position. We reference this figure in the Methods (Page 4, paragraph 1):

“Principal component analyses visualizing the ARIC and EHR populations with respect to HAPMAP populations are shown in Supplementary Figure 1 “

False discovery rate

To correct for multiple testing, the authors used a false discovery rate (FDR) of 0.1 to define significant biomarker-diagnosis associations. This means that ~38 of the 377 “significant” associations are expected to be false positives given a FDR of 0.1. Such a high number of false positives could lead to wrong conclusions about the genetic relationships between biomarkers and diagnoses. I would use a much more stringent FDR (or type-I error) threshold so that the number of false positives is smaller than 1 in the whole experiment.

The reviewer correctly notes that we would expect ~38 false positive associations at this threshold, which equates to $38/53=0.72$ false positive associations per biomarker analysis. Applying the reviewer’s criteria, we would need to apply a FDR cut-off of

~0.006, which identifies 117 significant associations with ~1 expected false-positive. We would argue that this approach is a misapplication of the FDR adjustment method since, for analyses such as these where many known true-positives associations are expected, forcing a lower FDR threshold to require a pre-specified number of false positives arbitrarily throws out true associations. In other words, to ensure that $(\text{FDR threshold}) \times (\text{number associations}) = 1$, the FDR threshold must be decreased based on the number of significant associations observed. We selected the 0.1 threshold because it is commonly used for PheWAS (e.g. see Denny JD et al. Nat Biotechnol. 2013 Dec;31[12]:1102-10), which is a hypothesis generating approach to identify candidate clinical diagnoses associated with genetic predictors. We would anticipate that any of the candidate associations we identify be replicated prior to implementation. For instance, we demonstrate that the LDL-septicemia association (which had an FDR value higher than that proposed by the reviewer) could be replicated epidemiologically. For all of the results that we report, we provide a table showing the FDR p-value so that readers can select subsets using alternative thresholds (see Supplementary table 7). We have modified the Methods as follows (Page 5, paragraph 4):

“Biomarker-phenotype pairs with FDR q-value<0.1 (a threshold commonly used for PheWAS⁹) were considered to have a statistically significant genetic association”

The total number of SNPs after HRC/1000G imputation should be >7 million. The authors reported that after LD reduction with a LD r-squared threshold of 0.9, there were ~740K SNPs. This number seems too small to me. Please clarify.

After imputation and data cleaning, the EHR data sets had 7,218,081 common SNPs available. The ARIC data set had 7,254,201 SNPs. The intersection of these data sets had 5,701,931 SNPs meeting QC criteria. The LD-reduced set had 739,681 SNPs. We have incorporated these counts in the Methods (Page 4, paragraph 3):

“The final analytic data set comprised a LD-reduced ($r\text{-square}=0.9$) set of 739,681 SNPs with $\text{MAF}>1\%$ present on all platforms (the EHR data sets had 7,218,081 SNPs, the ARIC data set had 7,254,201 SNPs and their intersection had 5,701,931 SNPs).”

Page 5. The equation used to compute the genetic predictor is not clear. Gamma is not defined. It seems that all the SNPs have an effect of alpha but only a subset of SNPs have an effect of beta with gamma being the mixing probability. The authors need to clearly define all the variables and parameters in the equation.

Per the suggestions of Reviewer #1, we have rewritten and clarified this section. Please see the response to Reviewer #1 above, which addresses these issues.

Page 5. I believe that the beta in the equation to compute skewness is not the same as the one used to compute the genetic predictor above. Please use a different notation.

The reviewer is correct that beta had 2 different meanings. As described in the response above, we have modified the Methods section so that the beta term is not used when describing the predictor.

Page 5. “from 0 (corresponding to a log(odds-ratio) of 1)” odds-ratio of 1?

The reviewer is correct. We removed the log term (Page 5, last paragraph):

“...from 0 (corresponding to an odds-ratio of 1)...”

Page 6. “For these analyses, eMERGE data were imputed using the Michigan Imputation Server”. Did the authors mean that non-EA samples were imputed to 1000G using the Michigan Imputation Server? There is no non-European sample in HRC.

The samples were imputed against the HRC (v1.1) reference using the Michigan Imputation Server. While enriched mostly for European ancestry, the HRC reference does include 1000 Genomes with diverse ancestry of 2495 samples. Hence, the HRC reference comprises a cosmopolitan population for the common genetic variation used to construct predictors in these analyses. We note that rare variation analyses would not be appropriate in this case.

Page 7. “an increased waist circumference increased the risk of the clinical diagnosis”. This sentence reads as if the association is causal.

We have modified this sentence to clarify that the analysis did not ascertain causality (Page 7, last paragraph):

“Each significant association had an odds-ratio >1 , indicating that a genetic predisposition towards an increased waist circumference was associated with an increased risk of the clinical diagnosis.”

Page 7. “The phenotype with next largest skewness value was LDL cholesterol, which was skewed in the opposite direction (skewness=-1.15).” It seems that LDL cholesterol is protective against most diagnoses. The authors might need to comment on this.

We have modified the Discussion to incorporate this finding and put it into context with other findings (Page 11, paragraph 2):

“For instance, higher levels of predicted LDL-C were associated with a diagnosis of hyperlipidemia and lower levels were associated with increased T2D risk, an association that has been previously reported.⁴⁵ Consistent with these findings, lowering cholesterol using a statin drug is associated with an increased risk of T2D.⁴⁶ This inverse genetic relationship was also manifest in the skewness analysis, which showed that the LDL-C predictor, which demonstrated the strongest negative skewing, had a skewness value of similar magnitude, but opposite direction, as a T2D predictor (-1.15 vs 0.98).”

Page 7. “In contrast, an alcohol predictor was associated with alcohol use, but not smoking (Supplementary figure 6).” This could just be a power issue.

This could be due to power. However, each predictor was applied to the exact same set of cases and controls, indicating that the alcohol predictor had a weaker association. We have modified the text to read (Page 8, paragraph 1):

“In contrast, an alcohol predictor was associated with alcohol use, but not smoking (Supplementary figure 8), indicating that the alcohol predictor was more weakly associated a smoking diagnosis.”

Page 8. “For 55 (14.5%) of the 377 significant associations, there is no clear or known genetic relationship between the biomarker and the associated diagnosis (Supplementary table 9).” It is difficult to know whether these are novel associations or false positives given the high false discovery rate.

As the reviewer noted earlier, among all of the associations, including those described in this section, ~10% are expected to be false positives based on the FDR selection criteria. From a statistical perspective, these associations are indistinguishable from the other 322 associations we report. For many of the results that we highlight in this

section, we discuss epidemiological and biological evidence supporting them in the Discussion section.

Page 8. “we ascertained how many associations observed in the EA analyses had a nominal association at $p < 0.05$ in a AA cohort.” The use of a p-value threshold of 0.05 is not justified. The authors need to apply a stringent threshold to correct for multiple testing.

To adjust for multiple testing, we have now applied a Bonferonni correction. We have also added text supporting the feasibility of this approach in AA populations. We have modified this section of the Results to read (Page 8, last paragraph):

“As described above, since we were insufficiently powered for discovery, we ascertained how many associations observed in the EA analyses had a Bonferonni-adjusted association at $p < (0.05/332 = 1.6 \times 10^{-4})$ in a AA cohort. Of the 377 EA associations, 322 were available for testing in the AA cohort. Seven associations in the AA cohort had a significant association. To ascertain whether the low replication rate could be due to low power, we compared the direction of association in the AA cohort to the direction in the EA cohort for all associations with a nominal association $p < 0.1$ in the AA cohort ($n = 86$). Among these, 81/86 (94.2%) had the same direction of association as the EA data set, suggesting that more associations would be replicated with a larger sample size.”

Page 9. “We found a novel inverse association between genetically predicted LDL-C levels and septicemia which was replicated in an independent EHR-derived epidemiological cohort.” The authors might also need to fit the first 20 PCs as covariates in the replication analysis.

As we clarified in response to comments from Reviewer #1 (above), this association used ungenotyped subjects and was evaluated using directly measured LDL-C levels. Hence, we cannot perform the PC adjustment.

The text in Figure 1B are difficult to read.

We have tried to enlarge the text, but are not able make the font larger without overlapping the labels and making them unreadable. We have generated a high resolution version of this image that can be enlarged to enable the labels to be easily read. We will include this figure with a final submission.

Figure 2A. It might be better to use a different color to represent the associations that fall in both “FDR < 0.1” and “Rank <= 5” categories.

We have used different colors to identify that additional categories proposed by the reviewer. We have also moved this figure to Supplementary figure 5, which allowed us to create a larger and easier to read display. A summary of the findings depicted in this supplementary figure is still shown in Figure 2.

Figure S1. “the proportion of variance proportion of genetic variance”. the proportion of genetic variance?

We have clarified this legend to read “...the histograms show the estimate of the proportion of phenotypic variance explained (PVE), the proportion of genetic variance explained by SNPs with large effects (PGE) and ...”.

Reviewers' comments:

Reviewer #1 (Remarks to the Author):

The authors have adequately addressed my comments.

Reviewer #2 (Remarks to the Author):

1. Population stratification

"Adjusting the analyses for up to 20 PCs only impacted one association, suggesting that the observed results were not due to population stratification (Supplementary figure 4)."

Since there is a strong outlier in Supplementary Figure 4, I think the main results presented in the paper should be based on the analyses fitting the first 20 PCs.

2. False discovery rate

"Biomarker-phenotype pairs with FDR q-value < 0.1 (a threshold commonly used for PheWAS9) were considered to have a statistically significant genetic association"

I would urge the authors to apply a stringent experimental-wise significance level to correct for multiple tests. The fact that applying a stringent threshold significantly reduces the number of significant discoveries is not a justification of using a less stringent threshold.

If the authors believe that a FDR threshold to control the number of false positive < 1 is a misapplication of the FDR method, they might need to use the Bonferroni method, which is the most widely used approach for multiple testing correction in GWAS.

This might also affect the conclusion that "For 55 (14.5%) of the 377 significant associations, there is no clear or known genetic relationship between the biomarker and the associated diagnosis (Supplementary table 9)."

3. Prediction equation

I queried the definition of the parameters in the prediction equation. The authors responded by removing the parameters and adding a vague statement that "The SNP weight (w) is the sum of the large and small effect sizes."

In the original version of the manuscript, there were three parameters in the prediction equation, i.e., alpha, beta and gamma. I had a read of the user manual of BSLMM. It says that alpha is the small (polygenic) effect, beta is the additional effect if the SNP belongs the large-effect group, and gamma is the posterior probability of the SNP being in the large-effect group. Therefore, the total effect size of a SNP = alpha + beta * gamma.

The authors need to clarify this in the manuscript rather than hiding the detail.

4. Imputation

"While enriched mostly for European ancestry, the HRC reference does include 1000 Genomes with diverse ancestry of 2495 samples."

I think the authors need to clarify this in the manuscript.

Comments to the editor

As requested, we provide detailed responses to the reviewer below. To ensure consistency and clarity, we also made a number of small edits, which are noted in the track changes document. We also moved the Methods section to the end of the manuscript per formatting requirements of the journal.

Responses to Reviewer #2

1. Population stratification

“Adjusting the analyses for up to 20 PCs only impacted one association, suggesting that the observed results were not due to population stratification (Supplementary figure 4).” Since there is a strong outlier in Supplementary Figure 4, I think the main results presented in the paper should be based on the analyses fitting the first 20 PCs.

We repeated all analyses adjusting for 20 PCs; however, for two biomarkers, the BSLMM program did not run to completion due to memory errors. Therefore, all of the analyses cannot be performed with adjustment for 20 PCs.

To more directly address the reviewer’s underlying concern, we note that we previously provided principal component plots showing that the study populations clustered tightly around European Ancestry populations (Supplementary Figure 13), consistent with the original selection of populations based on genetic ancestry. As the reviewer notes, when we repeated the analyses for the top associations and adjusted for 20 PCs, only 1 of 377 associations was not significant (i.e. the outlier we highlighted in Supplementary Figure 4). There are 2 important things to note about this single outlier. First, the association was between the biomarker ‘Red blood cell distribution width (RDW)’, a measure of the range of sizes of red blood cells, and a diagnosis of ‘Disorders of iron metabolism’ which are diseases affecting iron levels and typically present clinically with abnormal red blood cell sizes (which causes elevated RDW measurements). Thus, this represents a true positive association that was eliminated (e.g., becoming a false negative) with over-adjustment. Second, the RDW biomarker phenotype was only available on a relatively small number of individuals (~1,700), whereas the other biomarkers were measured in 5000-8,000 individuals. Thus, we believe that the most likely reason for the loss of significance of RDW result with 20 PCs was due to model overfitting. Overall, the data do not support the proposition that our findings are explained by population stratification. Incorporating 20 PCs results in failed analyses for 2 biomarkers and the loss of a probable true association for a third biomarker. We have clarified the outlying finding by modifying the Results section as follows (Page 4, Paragraph 1):

“To ensure that the associations were not due to population stratification, we re-ran the analyses for the top associations, adjusting for 20 PCs. Only one result was significantly impacted, an association between the biomarker ‘Red blood cell distribution width (RDW)’, a measure of the range of sizes of red blood cells, and a diagnosis of ‘Disorders of iron metabolism’ which are diseases affecting iron levels that often manifest with abnormal red blood cell sizes (Supplementary Figure 4). Unlike the other biomarkers, RDW was measured in a small number of subjects (n=1,736), so the attenuated association is likely due to the combination of the small sample size and model overfitting.”

2. False discovery rate

“Biomarker-phenotype pairs with FDR q-value<0.1 (a threshold commonly used for PheWASs) were considered to have a statistically significant genetic association.” I would urge the authors to apply a stringent experimental-wise significance level to correct for multiple tests. The fact that applying a stringent threshold significantly reduces the number of significant discoveries is not a justification of using a less stringent threshold. If the authors believe that a FDR threshold to control the number of false positive < 1 is a misapplication of the FDR method, they might need to use the

Bonferroni method, which is the most widely used approach for multiple testing correction in GWAS.

Per the reviewer's suggestion, we applied a Bonferroni adjustment in the primary analyses and identified 116 associations. We present this result in the revised Abstract and the Results section (Page 4, Paragraph 1):

"There were 116 biomarker-phenotype associations among 25 biomarkers that were significant at an experiment-wide Bonferroni $p < 0.05$ (Figure 1b and Supplementary Table 2). To ascertain the validity of the associations, we quantified how many of 42 pre-specified positive control biomarker-diagnosis pairs were significantly associated. Of 42 expected associations, 21 (50%) were significantly associated with a positive-control phenotype (Figure 2, Supplementary figure 3 and Supplementary table 3)."

We also modified multiple figures and tables to reflect these changes including Figures 1b, 2, supplementary figures 3, 7-11 and supplementary table 2.

The reviewer's suggestion also provided us an opportunity to compare selection using a Bonferroni selection threshold to an FDR selection adjustment, which we believe has enhanced the manuscript considerably. In the revised manuscript, we employ FDR as a discovery tool. We present new analyses comparing the results of the approaches (See new Figure 3 and Results under the heading "Evaluating an FDR-based selection threshold", Page 4, Paragraph 3). Of note, we show that selection based on FDR identifies more positive control associations and recapitulates more of the known epidemiology of the biomarkers. In subsequent sections in the Results, we highlight associations that meet both Bonferroni and FDR thresholds. However, we present results meeting each criteria in separate tables (Supplementary Table 2 and Supplementary Table 4), and have modified all figures to distinguish the associations meeting each threshold. We have also added a paragraph to the Discussion (Page 7, Paragraph 3):

"To assess the biological plausibility and validity of our approach, we examined 42 positive control associations in which the biomarker and the clinical phenotype were closely related. The positive control diagnosis was a top PheWAS association for 32 (76%) biomarker-diagnosis pairs, and was frequently the most significant association. Only 50% of the positive control associations were significant using a Bonferroni level of significance. PheWAS phenotypes are highly correlated, and a Bonferroni correction tends to be overly conservative since it assumes independence among tests.⁴⁶ The correlations among PheWAS phenotypes are almost exclusively positive and, thus, a Benjamini-Hochberg FDR adjustment is often applied, which accounts for the correlation structure among observations meeting the positive regression dependent criterion.^{11,47} For these analyses, in which many of the biomarkers are also closely related, accounting for correlation is important. We found that applying an FDR selection threshold identified more positive control associations (67%) and more known biological associations than a Bonferroni correction. PheWAS is typically used as a hypothesis generating tool, and the benefit of using an FDR threshold is that it identifies candidate associations while controlling for the proportion of false discoveries at a given selection threshold. Thus, at an FDR threshold of 0.1 used in these analyses, we identified far more candidate associations than when we applied a Bonferonni threshold. By design, ~10% of associations ($n \sim 38$) are expected to be false positives, so further validation of candidate associations, as we describe for LDL-C below, is essential."

This might also affect the conclusion that “For 55 (14.5%) of the 377 significant associations, there is no clear or known genetic relationship between the biomarker and the associated diagnosis (Supplementary table 9).”

We have removed this sentence and table in light of changes described above. Additionally, we felt the wording was confusing. Many of the associations referenced in the above statement have been consistently observed in epidemiological studies (such as an association between 1) white blood cell counts and bronchitis; 2) waist circumference and psoriasis; or 3) body mass index (BMI) and venous thrombosis). However, these phenotypes have not been associated with each other using a genetic approach. We removed this sentence because it gave the impression that many of these associations were likely false positives. Indeed, while this paper has been under review, genetic associations for several of these phenotypes have now been described (e.g. BMI and thrombosis [Circ Cardiovasc Genet. 2017 Apr;10(2)]; smoking and obesity [Nat Commun. 2017 Apr 26;8:14977]). We have retained the remainder of the paragraph, and we highlight interesting associations, which we further describe in the Discussion section.

3. Prediction equation

I queried the definition of the parameters in the prediction equation. The authors responded by removing the parameters and adding a vague statement that “The SNP weight (w) is the sum of the large and small effect sizes.” In the original version of the manuscript, there were three parameters in the prediction equation, i.e., alpha, beta and gamma. I had a read of the user manual of BSLMM. It says that alpha is the small (polygenic) effect, beta is the additional effect if the SNP belongs the large-effect group, and gamma is the posterior probability of the SNP being in the large-effect group. Therefore, the total effect size of a SNP = alpha + beta * gamma. The authors need to clarify this in the manuscript rather than hiding the detail.

We have modified the description for the computation of SNP weightings to provide the requested detail (Page 1, last paragraph):

“BSLMM was then used to generate SNP weights (w) using the regression residuals. For each SNP, BSLMM computes both a small polygenic effect (α), a large effect (β) and a posterior probability that the SNP is in the large effect group (γ) based on the underlying genetic architecture for the phenotype, as determined by the Bayesian algorithm. The SNP weight is computed using the equation: $w = \alpha + \beta\gamma$.”

Please note that, in response to a comment from the first round of revisions, we have substituted the symbol beta (β) for the word (“beta”) in the formula for skewness in order to avoid confusion with the beta term used above (Page 12, Paragraph 2).

4. Imputation

“While enriched mostly for European ancestry, the HRC reference does include 1000 Genomes with diverse ancestry of 2495 samples.” I think the authors need to clarify this in the manuscript.

We have modified the Methods to include the description of the HRC reference and cite the related publication (Page 12, Paragraph 3):

“This reference panel is enriched for individuals of European Ancestry and also includes the diverse ancestries from the 1000 Genomes populations ($n=2495$).²⁷”

Reviewers' comments:

Reviewer #2 (Remarks to the Author):

"Unlike the other biomarkers, RDW was measured in a small number of subjects ($n=1,736$), so the attenuated association is likely due to the combination of the small sample size and model over-fitting."

The number of observations ($n > 1700$) is still much larger than the number of variables ($n < 30$) fitted in the model. I don't see why there is an over-fitting problem.

"Benjamini-Hochberg FDR accounts for the correlation structure among observations meeting the positive regression dependent criterion.^{11,47} For these analyses, in which many of the biomarkers are also closely related, accounting for correlation is important."

It is incorrect to state that the Benjamini-Hochberg (BH) FDR approach accounts for the correlation structure among observations. I would urge the authors to have a read of the original BH paper (http://www.math.tau.ac.il/~ybenja/MyPapers/benjamini_yekutieli_A_NNSTAT2001.pdf). The BH approach is valid for independent observations and also valid in various scenarios of dependence. However, this doesn't mean that it accounts for the correlation structure. In fact, in data analysis, the only input are p-values which certainly do not contain any structure information.

Reviewer #3 (Remarks to the Author):

I looked over the paper and back and forth about Bonferroni vs FDR adjustment. I think it is prudent to use Bonferroni to report the main results and then FDR can be used describe potentially more interesting association that needs confirmation. I believe in the last revision that is what is the authors have done. So for me this is not a big concern.

I agree with the reviewer's comment also that it's not clear whether the BH procedure will be able to maintain the right false discovery rate given the dependence structure the data has. The positive control experiment, although interesting in terms of power, does not prove any evidence for the ability of the method to control type-I error.

I do have some additional concerns. Although I understand it may be late to ask the authors to address at this late stage, but I will put this out anyway

1) The authors have tried to sell the paper as if they are proposing a new design/paradigm for testing epidemiologic association between biomarkers and outcomes. Obviously, this assertion is not correct. There are many many studies which have already done this kind of analysis, sometimes under the rubric of Mendelian Randomization/Genetic Correlation analysis. The authors claim what they are doing something very distinct from MR analysis, but operationally it is not at all different - the question of whether and how one want to interpret the association in causal terms that is a separate issue.

This is not to say that the current data analysis is not interesting. It is useful to see results from a PheWAS in a large EHR using genetically predicted biomarkers.

2) For many if not most of of the "biomarkers" considered (e.g Blood Pressure, BMI, WHR, Cholesterol Traits), the ARIC study, which is a relatively small study in the standard of GWAS, is not the best study to build model for prediction. There are much larger publicly available data the authors could have used to build their models and that would have resulted in more powerful subsequent analysis in the EHR.

Responses to Reviewer #2

“Unlike the other biomarkers, RDW was measured in a small number of subjects (n=1,736), so the attenuated association is likely due to the combination of the small sample size and model over-fitting.” The number of observations (n > 1700) is still much larger than the number of variables (n < 30) fitted in the model. I don’t see why there is an over-fitting problem.

To address the reviewer’s concerns, we have revised this sentence to remove the suggestion of over-fitting (page 4, paragraph 1):

“Unlike the other biomarkers, RDW was measured in a small number of subjects (n=1,736), which may have contributed to its outlier status.”

“Benjamini-Hochberg FDR accounts for the correlation structure among observations meeting the positive regression dependent criterion.^{11,47} For these analyses, in which many of the biomarkers are also closely related, accounting for correlation is important.” It is incorrect to state that the Benjamini-Hochberg (BH) FDR approach accounts for the correlation structure among observations. I would urge the authors to have a read of the original BH paper (http://www.math.tau.ac.il/~ybenja/MyPapers/benjamini_yekutieli_ANNSTAT2001.pdf). The BH approach is valid for independent observations and also valid in various scenarios of dependence. However, this doesn’t mean that it accounts for the correlation structure. In fact, in data analysis, the only input are p-values which certainly do not contain any structure information.

We agree with the reviewer that our statement above is not accurate because it creates the impression that correlations within the data set are explicitly considered or modelled by the B-H FDR procedure.

Pertaining to the suitability of the FDR approach, we note that, in a clinical data set such as the one used in these analyses, the morbidity effects attributable to a biomarker may be distributed across a range of diagnoses. For instance, elevated systolic blood pressure manifests as headaches, strokes, subarachnoid hemorrhages, cardiomyopathies, a diagnosis of hypertension, kidney disease, peripheral vascular disease, etc. Some individuals will have multiple manifestations which, across the data set, can create positive correlations among these diagnoses. This distributed effect of a biomarker across a range of phenotypes causes clusters of related phenotypes to have associations with a biomarker in the same direction (thereby setting up positive regression dependence). In contrast, an absence of disease is not captured in the data set (for instance, there is not a diagnosis of “No elevated blood pressure”). Hence, inverse associations are not intrinsic to the data set. Given these features of the data and our stated goal of broadly identifying candidate clinical phenotypes associated with the biomarker, the FDR approach, which was designed to control the familywise error-rate in these settings, is the best tool. The appropriateness of FDR for these data is confirmed in the paper the reviewer references above: “When trying to use the FDR approach in practice, dependent test statistics are encountered more often than independent ones, the multiple endpoints example of the above being a case in point. A simulation study by Benjamini, Hochberg and Kling (1997) showed that the same procedure controls the FDR for equally positively correlated normally distributed (possibly Studentized) test statistics. The study also showed, as demonstrated above, that the gain in power is large.”

We have rewritten this Discussion (and replaced the text highlighted by the reviewer) to more accurately describe the utility of the FDR for these data (page 7, paragraph 3):

“Because the effects of a disease process can manifest across a range of phenotypes, a disease biomarker will often be associated with multiple diagnoses. In these instances, the FDR adjustment procedure is desirable since it is designed to control the familywise error-rate in the context of positive regression dependence.”^{11,32}

We have also modified the limitations section of the Discussion to note the assumptions of the FDR approach (page 9, paragraph 3):

“The B-H FDR method used in these analyses may have an erroneous type 1 error rate if the pattern of associations violates the positive regression dependence assumption.”

To empirically assess whether the type I error rate of the FDR procedure is large, we evaluated permuted data. We permuted the output from the BSLMM algorithm and applied the FDR selection threshold to 1000 sets of 53 permuted predicted biomarkers values. Importantly, this strategy preserves the relationships among the clinical phenotypes. In 86 permutations (~8.6%), there were one or more significant associations observed (FDR $q < 0.1$), and no permutations had more than 3 associations. Eight (0.8%) permutations had either 2 or 3 associations. These results indicate that the FDR procedure is performing as expected and is not inflating type 1 error in this data set. In prior a revision of this manuscript, the reviewer stipulated that “I would use a much more stringent FDR (or type-I error) threshold so that the number of false positives is smaller than 1 in the whole experiment.” The analyses suggest that these criteria have been met, as there is an ~0.8% chance that there will be more than 1 false positive result for the entire experiment under the null hypothesis.

Responses to Reviewer #3

I looked over the paper and back and forth about Bonferroni vs FDR adjustment. I think it is prudent to use Bonferroni to report the main results and then FDR can be used describe potentially more interesting association that needs confirmation. I believe in the last revision that is what is the authors have done. So for me this is not a big concern.

This was, indeed, the approach that we took for the analyses. Consistent with this, the only specific finding we report in the Abstract was an association that we validated in the manuscript.

I agree with the reviewer's comment also that it's not clear whether the BH procedure will be able to maintain the right false discovery rate given the dependence structure the data has. The positive control experiment, although interesting in terms of power, does not prove any evidence for the ability of the method to control type-I error.

Please see our responses to the reviewer above regarding use of the B-H procedure.

We agree that the positive-control experiment does not address the issue pertaining to control of type I error. The purpose of those analyses was to leverage known knowledge in order to construct a principled, biologically-motivated framework to analyze and characterize our findings.

1) The authors have tried to sell the paper as if they are proposing a new design/paradigm for testing epidemiologic association between biomarkers and outcomes. Obviously, this assertion is not correct. There are many studies which have already done this kind of analysis, sometimes under the rubric of Mendelian Randomization/Genetic Correlation analysis. The authors claim what they are doing something very distinct from MR analysis, but operationally it is not at all different - the question of whether and how one want to interpret the association in causal terms that is a separate issue. This is not to say that the current data analysis is not interesting. It is useful to see results from a PheWAS in a large EHR using genetically predicted biomarkers.

As the reviewer notes, we state that “These analyses are distinct from a Mendelian Randomization design, which seeks to establish a causal role between a phenotype and an outcome.” The purpose of this comment was to emphasize that we do not propose causality between an associated biomarker and clinical diagnoses. We have now modified this sentence to more clearly emphasize this point and to ensure that it does not convey the inaccurate impression that a genetic association approach is unique to this manuscript (Page 9, Line 3).

“These analyses do not meet the criteria of a Mendelian Randomization experiment, a subclass of genetic association studies which seek to establish a causal association between a biomarker and an outcome.^{5,47} Thus, similar to an epidemiological or genetic correlation study, we also do not claim causality for the associations that we report.”

2) For many if not most of of the "biomarkers" considered (e.g Blood Pressure, BMI, WHR, Cholesterol Traits), the ARIC study, which is a relatively small study in the standard of GWAS, is not the best study to build model for prediction. There are much larger publicly available data the authors could have used to build their models and that would have resulted in more powerful subsequent analysis in the EHR.

We agree that larger data sets would have enabled better genetic instruments to be developed, as has been reported. However, as described in the Introduction, our goal for these analyses was to ascertain whether integrating a Bayesian linear modelling approach with phenome-wide scanning could rapidly delineate the clinical disease spectrum associated with a biomarker that was measured in a relatively small data set, which is the setting where a novel/unproven biomarker is typically measured. Hence, the selection of the ARIC data set was intentional, as it represents a real-world example of a data set where novel biomarkers are often measured and evaluated.

REVIEWERS' COMMENTS:

Reviewer #2 (Remarks to the Author):

The authors have addressed all my concerns. I have no further comment.

REVIEWERS' COMMENTS:

Reviewer #2 (Remarks to the Author):

The authors have addressed all my concerns. I have no further comment.
We appreciate the input from the reviewer.